# MIDGArD: Modular Interpretable Diffusion over Graphs for Articulated Designs

Quentin Leboutet     Nina Wiedemann     Zhipeng Cai     Michael Paulitsch     Kai Yuan

Intel Labs – XRL – eXtended Reality Laboratory
{firstname.lastname}@intel.com

## Abstract

Providing functionality through articulation and interaction with objects is a key objective in 3D generation. We introduce MIDGArD (Modular Interpretable Diffusion over Graphs for Articulated Designs), a novel diffusion-based framework for articulated 3D asset generation. MIDGArD improves over foundational work in the field by enhancing quality, consistency, and controllability in the generation process. This is achieved through MIDGArD's modular approach that separates the problem into two primary components: *structure generation* and *shape generation*. The structure generation module of MIDGArD aims at producing coherent articulation features from noisy or incomplete inputs. It acts on the object's structural and kinematic attributes, represented as features of a graph that are being progressively denoised to issue coherent and interpretable articulation solutions. This denoised graph then serves as an advanced conditioning mechanism for the shape generation module, a 3D generative model that populates each link of the articulated structure with consistent 3D meshes. Experiments show the superiority of MIDGArD on the quality, consistency, and interpretability of the generated assets. Importantly, the generated models are fully simulatable, i.e., can be seamlessly integrated into standard physics engines such as MuJoCo, broadening MIDGArD's applicability to fields such as digital content creation, meta realities, and robotics.

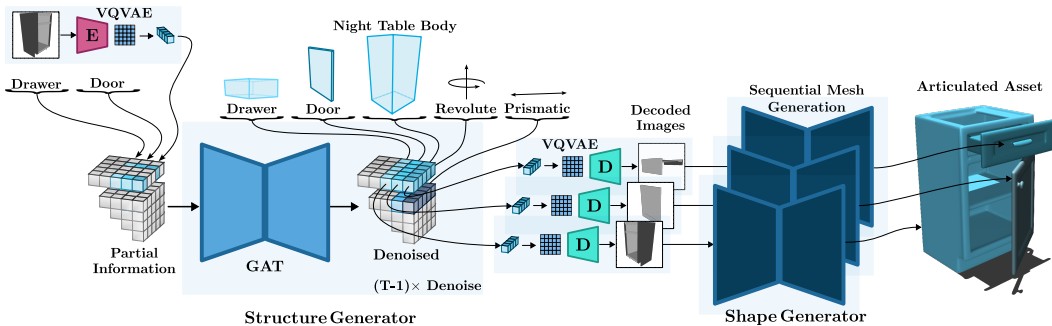

Figure 1: The MIDGArD generative pipeline.

## 1   Introduction

Despite their significance for applications in meta realities and embodied AI, the creation of 3D models of articulated objects remains a manual endeavor, and existing datasets [92, 97, 65, 102, 47, 63, 15] are often limited in both scope and scale. Human designers utilize strong prior knowledge of object geometry and kinematic structures, a capability that automated systems have yet to fully replicate.

38th Conference on Neural Information Processing Systems (NeurIPS 2024).

While extensive research has been conducted on generative models for static 3D objects [78, 74, 50, 48, 35, 8, 39, 54, 2, 95, 103, 24, 68, 9, 88, 75, 93, 6, 85, 98, 77, 52] and scenes [38, 31, 17, 71, 84, 14, 72, 53, 7, 12, 107], integrating 3D part geometry with kinematic structures has received far less attention. Generating articulated objects that are both geometrically detailed and kinematically accurate presents unique challenges due to the complexity of modeling part interactions and motions. Recent research efforts, such as Neural 3D Articulation Prior (NAP) [37] or Controllable Articulation GEneration (CAGE) [46], started addressing this gap, pioneering direct end-to-end generation of complete articulated assets using a denoising diffusion process acting on articulation graphs. However, generating consistent objects with high-fidelity shapes remains a challenge. Identified shortcomings of NAP [37] include 1) unnatural motion caused by unconstrained screw-like joint parametrization, 2) limited controllability, 3) limited interpretability and 4) the generation of inconsistent or unidentifiable shapes. Additionally, both NAP and related approaches lack *part*-level control due to the opaque encoding as a latent within the graph, highlighting the need for enhanced interpretability.

In this work, we introduce MIDGArD, a novel framework designed to generate interpretable and simulatable 3D articulated assets in a controllable manner. MIDGArD addresses the limitations of previous methods by offering: 1) improved consistency in joint motion, 2) enhanced controllability of the generation process, 3) increased interpretability and 4) better overall asset quality. Furthermore, the 3D assets generated by MIDGArD are fully simulatable and can be seamlessly integrated into virtual environments or physics engines such as MuJoCo [86], significantly enhancing its utility for digital content creation and robotics. MIDGArD employs a modular and sequential approach based on two diffusion models: the *structure generator* and the *shape generator* (see Figure 1). The *structure generator* denoises an articulation graph, unconditionally or from incomplete heterogenous inputs. This articulation graph acts as an abstract, yet interpretable object representation, encoding the structural and semantic information of every link, as well as kinematic attributes.

MIDGArD overcomes the identified limitations of prior works by 1) leveraging categorical embeddings for improving consistency across the articulation graph; 2) representing nodes (i.e. unique object parts) via latent image codes rather than direct mesh encoding; 3) diffusing on the Plücker manifold, thereby eliminating the need for discretizing the generated motion parameters, and 4) accounting for the orientation of the parts for more consistent generations. As additional benefit, these modifications yields an image representation, and asset-level, body-level, and joint-level categorical data, that are *human interpretable*. Since the generation process is sequential, these quantities can be adjusted by the user before generating proceeding with the part geometry generation. Crucially, this interpretable structure enables the use of a multi-modal 3D generation model as the *shape generator*. The shape generator, here represented by SDFusion [8], is queried multiple times to produce high-quality geometries for each part of the object (see Figure 1). To enhance consistency, we propose a constrained generation approach that enforces the part geometry to fit a given bounding box, here provided by the structure generator.

We demonstrate MIDGArD's capabilities by showcasing improved structural quality, consistency, and part-level control compared to NAP. Furthermore, we show qualitative results of MIDGArD generating diverse articulated objects dependent on the partial graph input and shape conditioning. To further improve the consistency and quality of the generated shapes, we propose a to create oriented bounding boxes that have a tighter fit to the ground truth while being properly aligned. The reduces the overlapping volume by 17% on average. Our contributions can be summarized as:

1. **MIDGArD:** A novel diffusion-based framework for generating high-quality, articulated, and simulatable 3D assets with improved consistency between structure and shape.

2. **Enhanced Interpretability, and Controllability:** Providing human-interpretable intermediate outputs and allowing user adjustments at multiple levels of the generation process.

3. **Enhanced Quality and Scalability:** Relying on state-of-the-art 3D foundation models for shape generation.

4. **Improved part alignment:** Introducing a bounding-box constrained shape generation approach that improves alignment of kinematic links by 17% on average.

5. **Simulatable Assets:** Offering a pipeline to create fully simulatable assets compatible with physics engines such as MuJoCo.

Code and models are available at https://quentin-leboutet.github.io/MIDGArD.

## 2 Related Works

**Structural Characterization and State Estimation of Articulated Objects**    Designing articulated objects involves understanding both their geometry and kinematic properties. Traditional methods often require extensive manual labor or rely on limited datasets, hindering scalability and applicability [45]. Early work on morphable templates, such as SMPL for human bodies [55] and SMAL for quadruped animals [114], demonstrated the potential of matching existing articulation structures (i.e., skeletons) to a wide variety of inputs such as static meshes or monocular videos, thereby substantially simplifying the rigging process. However, while effective within specific categories, these templates lack the flexibility to generalize to arbitrary articulated mechanisms. To address these limitations, recent approaches have enhanced templates by incorporating implicit representations to model geometric deformations and appearances [40]. Robust neural rigging methods [61, 70] leverage template priors while remaining flexible enough to handle mesh elements that deviate from the template, such as clothing or hair. On the other hand, template-free methods [100, 101] rely solely on input mesh geometry but are still primarily limited to humanoid characters due to the scarcity of comprehensive datasets. Emerging approaches [27] aim to rig arbitrary static assets, potentially paving the way for a unified, template-free rigging pipeline. Our research aligns with this direction by learning articulation priors for arbitrary objects without depending on predefined templates.

Significant strides has also been made in estimating articulation joint states and parameters from sensory observations. Early contributions such as [83] demonstrated that it is possible to learn compact kinematic models of entire articulated objects solely based on pose observations. Probabilistic models have enabled robots to learn their own kinematic structures through self-observation and motor babbling [10, 81, 82]. Gaussian processes have been employed to model part connectivity and link articulation in objects [87]. Interactive perception techniques further enhance this learning process by allowing robots to interact with objects to obtain their kinematic model [69, 33, 64, 18, 66, 3, 22, 44, 25]. Deep learning techniques were also used to train dedicated architectures to directly predict articulation models from sensory inputs [23, 104, 1, 41, 49, 94, 106, 34, 29, 28, 51, 102, 13, 99, 105, 43, 20, 109]. Interactive systems [67, 13] learn to predict potential motions of articulated objects, aiding downstream motion planning and interaction, while reinforcement learning based approaches [96] train policies to explore diverse interaction trajectories, contributing to actionable visual representations. Our method diverges from traditional approaches by not relying on explicit estimation pipelines; instead, we learn articulation priors implicitly within our model.

**Articulated Object Generation**    Generating articulated objects involves synthesizing models that accurately represent both geometric details and kinematic behaviors. Methods such as Self-Supervised Category-Level Articulated Object Pose Estimation [51] and PD (Pose-aware Part Decomposition) [34] address the generation of joint parameters and part poses, respectively, contributing significantly to the field. Few generative frameworks address both aspects concurrently. Notable among them is NAP [37], which generates complete articulated assets—including meshes—using graph attention networks. CAGE [46], enhances controllability and interpretability, albeit with limitations in mesh generation. More recently, [62] proposed a framework capable of reconstructing articulated assets from image inputs by using a 3D shape completion model to generate the different parts and a large language model (LLM) to predict the joint parameters. Progress was also made in physics-aware shape generation; for instance, [58] introduces an improved loss mechanism to account for internal collisions within generated articulated assets. In this work, we build up on NAP and CAGE and design an approach to enhance the controllability of the generative process, and leverage a multi-modal 3D generative model for improving the quality of the shape geometry.

**3D Shape Generation and Completion**    Recent progress in diffusion models and 2D-guided generation techniques have revolutionized 3D shape generation and completion. Two main types of generative pipelines are currently being explored. The first one leverages 2D diffusion models to optimize NeRFs [73, 42, 57], enabling high-quality 3D generation without the need for direct 3D supervision. The second employs Latent Diffusion Models (LDMs) for different 3D representations, including point clouds [59, 113, 112, 60, 88], voxels [111], SDFs[8, 76, 111, 48], shape2vecset[108, 110], and triplanes [16]. It should be emphasized that despite the predominant role of diffusion models, other methodologies such as autoregressive models [78], have demonstrated potential in producing high-fidelity 3D shapes from minimal inputs. Conditional diffusion models enable multi-modal inputs. SDFusion [8], for instance, can generate 3D objects from text descriptions, images, or partial 3D shapes. Building upon these capabilities, we have adapted SDFusion as the shape generator within our framework. However, due to MIDGArD's modular design, the shape generator can be updated to incorporate the latest advancements in static 3D generation.

# 3 Synthesis of 3D Articulated Mechanisms

## 3.1 Method Overview

Both components of MIDGArD, namely the structure and the shape generator, leverage a denoising diffusion model [79, 80, 21]. Diffusion models operate by progressively degrading training data through the systematic application of Gaussian noise, a method known as the forward process. Subsequently, these models are trained to restore the original data by methodically removing this noise in what is known as the reverse process. Please refer to Appendix A and [21], [79] for details. As shown in Figure 1, the full MIDGArD pipeline can be separated into the *Structure Generator* and the *Shape Generator*. The Structure Generator (subsection 3.2) takes graph-based representation of an articulated asset – which may be incomplete or affected by noise – and resolves this representation through diffusion. The denoised graph features, decoded into suitable – human interpretable – text prompts, images and bounding box information are then fed into the Shape Generator (subsection 3.3), another diffusion-based model yielding the final articulated 3D object.

## 3.2 Structure Generator

**Articulated Asset Representation and Parametrization**  Building upon [37], we encode the structural and kinematic attributes of every articulated asset into the node and edge features of a complete graph referred to as $\boldsymbol{G}_N$. Our structure generation module leverages a Graph ATtention (GAT) [90, 4, 32] denoising network to generate coherent and interpretable articulation features from noisy or incomplete inputs (see Figure 1). To improve the quality, controllability, and interpretability compared to [37], we modify the asset parametrization. Drawing on insights from [46], we incorporate a set of categorical variables to the nodes and edges features to enhance asset consistency, interpretability and provide a more intuitive control interface over the generation process. Each asset is represented as a complete graph $\boldsymbol{G}_N = \{\boldsymbol{x}, \boldsymbol{e}\}_N$ that embodies the object's links and joints in its $N$ nodes, and in its $N(N-1)/2$ edges respectively. Rather than directly denoising and then decoding a low-dimensional shape latent $\boldsymbol{f}_i \in \mathbb{R}^{D_F}$ for each link $i$ into a 3D mesh [37, 36], we propose a two-step approach that operates on a more manageable image latent $\boldsymbol{g}_i \in \mathbb{R}^{D_G}$ of the link. This image latent is derived by training a Vector-Quantized Variational Auto Encoder (VQVAE) [89] on various views of every rigid body in the dataset. In our method, the structure generator denoises a latent representation of an image for each individual component of the articulated asset. This denoised image, along with the categorical information extracted from the graph, facilitates the creation of human-interpretable priors for each link, namely a text description and a front view image. These prompts can then be utilized to condition a 3D shape generation model yielding consistent meshes. Additionally, the graph itself encodes highly relevant structural features that act as supplementary conditioning signals, further guiding the shape generation model.

Each node $\boldsymbol{x}_i$ represents a link of the articulated asset and $\forall i, j \in [0, N-1]$ the edge $\boldsymbol{e}_{ij}$ establishes a connection between node $\boldsymbol{x}_i$ and node $\boldsymbol{x}_j$. The node feature vector $\boldsymbol{x}_i$ is comprised of multiple components, specifically, $\boldsymbol{x}_i = [\boldsymbol{o}_i, \boldsymbol{a}_i, \boldsymbol{b}_i, \boldsymbol{d}_i, \boldsymbol{r}_i, \boldsymbol{t}_i, \boldsymbol{g}_i] \in \mathbb{R}^{D_V}$, and the edge feature vector $\boldsymbol{e}_{ij}$ is formulated as $\boldsymbol{e}_{ij} = [\boldsymbol{c}_{ij}, \boldsymbol{j}_{ij}, \boldsymbol{p}_{ij}, \boldsymbol{l}_{ij}] \in \mathbb{R}^{D_E}$, where $D_V$ and $D_E$ refer to the node and edge feature dimensions. To ensure a consistent graph size across various objects, $N$ is set to the maximum count of parts anticipated, using $\boldsymbol{o}_i \in [0, 1]$ as a binary indicator to denote part existence. Similarly, the edges denote symbolic existence and chirality through the indicator variable $\boldsymbol{c}_{ij} \in [0, 1]^3$. Joint types are represented through $\boldsymbol{j}_{ij} \in [0, 1]^{D_J}$, a one-hot encoding that in our design has a dimensionality of three to signify the types – prismatic, revolute, or screw. Similarly, $\boldsymbol{a}_i \in [0, 1]^{D_A}$ and $\boldsymbol{b}_i \in [0, 1]^{D_B}$ denote the one-hot encoding vectors of asset (resp. body) categories. Similar to [37], we use Plücker coordinates as joint parametrization as this provides a compact representation for joint types, such as revolute, prismatic, and screw joints. The Plücker joint parametrization, articulated as $\boldsymbol{p}_{ij} \in \mathbb{R}^6$, provides a harmonious portrayal of the different joint types under consideration, and joint limits are entailed by $\boldsymbol{l}_{ij} \in \mathbb{R}^{D_L}$. Ultimately, the vectors $\boldsymbol{d}_i \in \mathbb{R}^3$, $\boldsymbol{r}_i \in \mathbb{R}^3$ and $\boldsymbol{t}_i \in \mathbb{R}^3$ depict the bounding box dimensions, orientation and position of each part. Note that $\boldsymbol{r}_i$ and $\boldsymbol{t}_i$ are defined for every part relative to its parent reference frame.

Unlike previous approaches [37, 46], we do not assume that the parts of the dataset are provided in a canonical orientation. Assuming that objects are in canonical orientation and not explicitly considering the orientation leads to degrading performance due to misaligned orientations of the dataset (see Figure 5, as well as Appendix B-Figure 10 and Appendix B-Figure 11). Instead, we

estimate this orientation for each body in the dataset by computing the corresponding Oriented Bounding Box (OBB), which is then used to position the body in a canonical pose. The process (see Appendix B) begins by generating a convex hull from the mesh. This step simplifies the complex geometry into a more manageable form, reducing dependency on the internal structure of the object. Next, we voxelize the convex hull to collect volumetric samples. We then execute a Principal Component Analysis (PCA) to determine the primary axes of the mesh providing an initial approximation of the object's orientation. Following the PCA, we perform gradient descent on the attitude quaternion of the OBB, using the PCA-based initial estimate as a starting point. The objective of this optimization is to minimize the volume of the OBB. This approach allows for a more flexible and accurate alignment of dataset bodies into canonical poses, thereby enhancing the overall quality and consistency of the articulated objects generated by our framework (see Figure 5, Table 3).

**Denoising Diffusion on the Plücker Manifold** To maintain the interpretability of the denoised quantity as a Plücker vector $\boldsymbol{p}_{ij} = [\boldsymbol{u}_{ij}, \boldsymbol{v}_{ij}]$, our approach introduces a novel parametrization allowing the diffusion process to be directly executed on the Plücker manifold $\mathcal{P}$. In contrast, the approach proposed in [37], denoises within $\mathbb{R}^6$, thereby requiring a projection operation $p : \mathbb{R}^6 \to \mathcal{P}$ at every denoising step. We propose an alternative formulation $\boldsymbol{k}_{ij} = [\boldsymbol{m}_{ij}, \boldsymbol{n}_{ij}] \in \mathbb{R}^5$ of the joint parameters where $\boldsymbol{m}_{ij} = [\theta_{ij}, \varphi_{ij}] \in \mathbb{R}^2$ represent the spherical parametrization of the Plücker axis $\boldsymbol{u}_{ij}$. This formulation yields $\boldsymbol{u}_{ij} = [\sin(\theta_{ij})\cos(\varphi_{ij}), \sin(\theta_{ij})\sin(\varphi_{ij}), \cos(\theta_{ij})] \in \mathbb{R}^3$ with $\|\boldsymbol{u}_{ij}\| = 1$. The vector $\boldsymbol{n}_{ij} \in \mathbb{R}^3$ is then defined such that $\boldsymbol{v}_{ij} = \boldsymbol{n}_{ij} \times \boldsymbol{u}_{ij}$. This alternative parametrization guarantees the normalization of $\boldsymbol{u}_{ij}$ and enforces by construction the orthogonality between $\boldsymbol{u}_{ij}$ and $\boldsymbol{v}_{ij}$, which are key features of the Plücker manifold [30]. Consequently, our approach inherently satisfies the manifold's constraints, thereby streamlining the computational workflow by eliminating the necessity for iterative projections.

### 3.3 Shape Generator

The shape generator of MIDGArD aims to create the mesh for each component of an articulated object using the denoised object graph, which contains both semantic (object category) and visual (image latents) data. We leverage a multi-modal 3D generative model that is trained independently from the structure generator. Our modular setup is designed to accommodate any generative model, allowing for seamless integration of the latest advances in 3D generation. In this work, the SDFusion model [8] is used, conditioned on image, text as well as graph features input, and enhanced with a novel approach for bounding-box-constrained generation. The training pipeline is shown in Figure 2.

**Multimodal Guidance for Consistent Shape Generation** The core of SDFusion is a latent diffusion model, see [8] for details. Object meshes are transformed into truncated signed distance functions (TSDFs) and encoded using a pretrained and frozen 3D VQ-VAE [89] model. The diffused latent representations, along with image and text condition signals, are denoised through a 3D U-Net model. The resulting output latent is then decoded into a TSDF and converted back into a mesh using a marching cube algorithm [56]. It is worth emphasizing that we train and apply the diffusion model on the individual *parts* of each objects, querying the model multiple times at inference time to generate a full articulated object (see Figure 1). The model is trained on the PartNet Mobility dataset [97] with the same train-test split as the structure generator. We modified SDFusion's conditioning [8] to align it with the output of the structure generator. The model is conditioned through cross-attention on the following modalities, either independently or in combination.

- Single view image: A pretrained ResNet-18 model [19] encodes images. During training, the model uses renderings of the part mesh from a frontal view. At inference, the model decodes the image from the node features of the articulation graph.

- Text: We employ BERT [11] for text encoding (similar to [84, 8]). The text is constructed using the object category and asset type from the PartNet dataset in the format "A `<asset type>` as part of a `<object category>`"; for instance, "*A lid as part of a trash can*".

- Graph: The output from the structure generator is encoded and concatenated with image- and text embeddings to provide information about the part's role within the object, such as its expected size and its relation to other parts, i.e. joint types.

This multi-modal conditioning approach ensures the generation of consistent, high-quality parts that can be seamlessly integrated to form complete articulated objects, enhancing the overall performance and applicability of our framework.

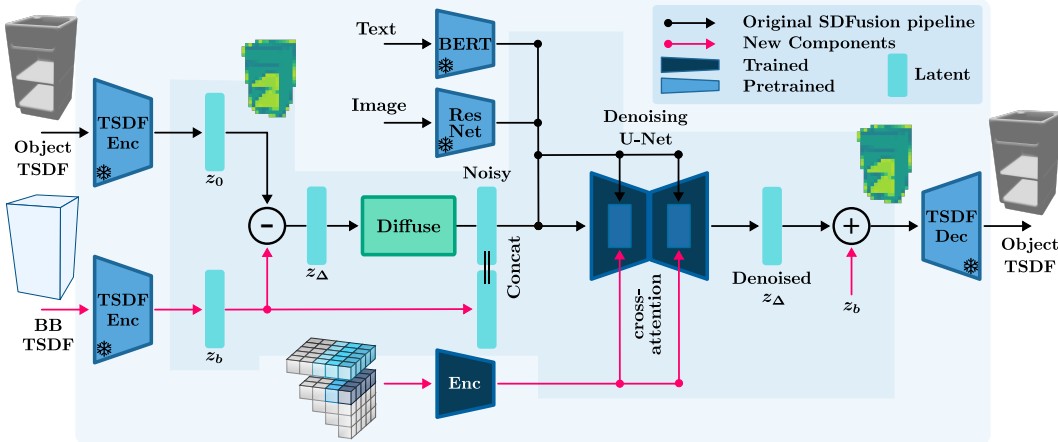

Figure 2: Architecture of the Shape Generator.

**Enforcing Bounding Constraints**    Generating object parts independently offers increased flexibility and quality but introduces challenges related to the size and orientation of the generated parts. Intuitively, the shape generator must ensure that parts conform to $r_i$ and $d_i$ in the articulation graph. To achieve the correct *orientation*, we generate links in a canonical pose and subsequently apply a post-processing rotation based on $r_i$, the 3D orientation vector derived during structure generation. To train the model accordingly, by estimating and adjusting the pose of parts to a canonical orientation, as detailed in Appendix B.

To ensure that generated parts match the *sizes* specified within the graph node features, we introduce a bounding-box constraint to the generative model (see Figure 2). Instead of generating the object directly, the diffusion model is trained on the *difference* relative to a predefined bounding box. To enable this schema for *latent* diffusion, we calculate the difference as $z_\Delta = z_o - z_b$, where $z_o$ is the VQ-VAE-encoded latent of the object, and $z_b$ is the encoding of a simplified bounding-box TSDF (values inside the box set to -0.01, and values outside set to 0.01). During inference, we provide the denoiser with the bounding box information by concatenating $z_b$ to the noisy version of $z_\Delta$. Other conditioning inputs, such as images, text, and graph features encodings, are incorporated through cross-attention mechanisms. The object's TSDF is then computed by decoding $z_\Delta + z_b$ using the 3D VQ-VAE. This method effectively guides the model to produce objects that are closely matching the target bounding boxes even after a few training steps.

## 4    Experiments, Results and Discussion

### 4.1    Experiment Setup

**Dataset**    All experiments were conducted using the PartNet Mobility dataset [97], which contains a diverse set of articulated 3D objects with detailed geometric and kinematic annotations. Each mesh in the dataset underwent a two-step preprocessing routine. First, we enforced the manifold property as described in [26]. Second, we performed orientation estimation to establish a canonical pose for each object and accurately compute the corresponding bounding boxes (see Appendix B for details).

**Evaluation Metrics**    We adopt the evaluation framework from NAP [37], which introduces the Instantiation Distance (ID) metric for comparing two articulated objects. ID measures the average Chamfer distance over sampled joint states, accounting for both geometric and kinematic differences. It is defined as:

$$ID(O_1, O_2) \approx \frac{1}{M} \sum_{q_1 \in Q_1} \left[ \min_{q_2 \in Q_2} \left( \tilde{d}(O_1, q_1, O_2, q_2) \right) \right] + \frac{1}{M} \sum_{q_2 \in Q_2} \left[ \min_{q_1 \in Q_1} \left( \tilde{d}(O_1, q_1, O_2, q_2) \right) \right], \quad (1)$$

where $Q$ is a set of $M$ uniformly sampled joint poses and $\tilde{d}$ is the minimum Chamfer distance over all possible canonicalizations of the mesh. We use ID in conjunction with standard metrics for unconditional generation, namely minimum matching distance (MMD), coverage (COV) and 1-nearest-neighbor accuracy (1-NNA).

**Implementation Details**    We trained the structure generator and the image VQ-VAE on an NVIDIA RTX 3090 GPU, while the shape generator was trained on an NVIDIA RTX 6000 GPU. Evaluation took place on a single NVIDIA RTX 3090 GPU. The image VQ-VAE, which encodes latent representations of objects in the shape generator, was trained on $256 \times 256$-pixel front renderings of the the PartNet Mobility object meshes. The denoising model used in the structure generator contains six graph attention blocks, with a latent embedding size of 512 and 32 attention heads. We set the maximum number of nodes in the graph to $N = 8$. Our training parameters closely follow those in NAP, with the key difference being the use of an implicit denoising diffusion pipeline [80] over 100 time steps, as opposed to a DDPM with 1,000 time steps. This modification significantly accelerates inference speed. Our shape generator is adapted from SDFusion [8] and trained on the PartNet Mobility dataset. We used the same hyperparameters as the multimodal model in SDFusion and utilized their pre-trained VQ-VAE checkpoint. We excluded 10 categories from training due to their objects containing numerous equally-shaped parts (e.g., keyboards with over 30 keys).

## 4.2    Results and Discussion

**Unconditional Articulated Asset Generation**    To the best of our knowledge, NAP is the only approach for generating articulated 3D objects without prior knowledge about their geometric structure. We first evaluate MIDGArD in an unconditional generation setting, comparing its performance to NAP. For a fair comparison, we retrained NAP on our preprocessed version of the PartNet Mobility dataset. The results are presented in Table 1. MIDGArD outperforms NAP in terms of MMD and COV metrics, indicating better diversity and coverage of the generated samples. Specifically, we observe a 6.4% improvement in MMD and a 3.7% improvement in COV. The 1-NNA metric is comparable between both methods. We also perform an ablation study on the effect of the Plücker manifold parameterization. As shown in Table 1 using the Plücker manifold improves the MMD and COV metrics, suggesting enhanced consistency and diversity in the generated assets.

Table 1: Comparison to NAP in an unconditional generation setting.

| Generative Paradigm/Method | Unconditional ID | | |
| --- | --- | --- | --- |
| | MMD ↓ | COV ↑ | 1-NNA ↓ |
| NAP | 0.0282 | 0.4675 | **0.5831** |
| Ours (Plücker manifold) | **0.0264** | **0.4857** | **0.5831** |
| Ours (No Plücker manifold) | 0.0270 | 0.4779 | 0.6221 |

To further evaluate the physical plausibility of the generated assets, we analyze the distribution of joint types in the training data and compare it with those in samples generated by NAP and MIDGArD. We sampled 400 objects generated by each method and counted each joint type only once per object to minimize the impact of objects with multiple joints of the same type.

Table 2: Distribution of joint categories in real data vs generated data from NAP and our approach.

| | Screw | Revolute | Prismatic |
| --- | --- | --- | --- |
| Training data | 6% | 62% | 32% |
| NAP-generated | 95% | 1% | 4% |
| MIDGArD-generated | **2%** | **62%** | **36%** |

The results in Table 2 show that NAP predominantly produces screw joints, which are rare in the training data. In contrast, MIDGArD generates objects with a joint type distribution closely matching that of the training data, thereby enhancing the plausibility of the generated assets. We computed the Chi-Square statistic to quantify the deviation from the expected joint type distribution. NAP's generated data yielded a high $\chi^2$ value of 5618, indicating a significant difference from the training data distribution. In contrast, MIDGArD's generated data resulted in a low $\chi^2$ value of 12.7, demonstrating close alignment with the expected distribution.

Figure 3.B provides a side-by-side qualitative comparison between MIDGArD and NAP. MIDGArD generates objects with higher geometric quality and more realistic motion compared to NAP. For instance, the fan generated by MIDGArD exhibits detailed geometry, and the laptop displays realistic opening motion, whereas NAP's outputs often show unnatural joint motions and inconsistent shapes.

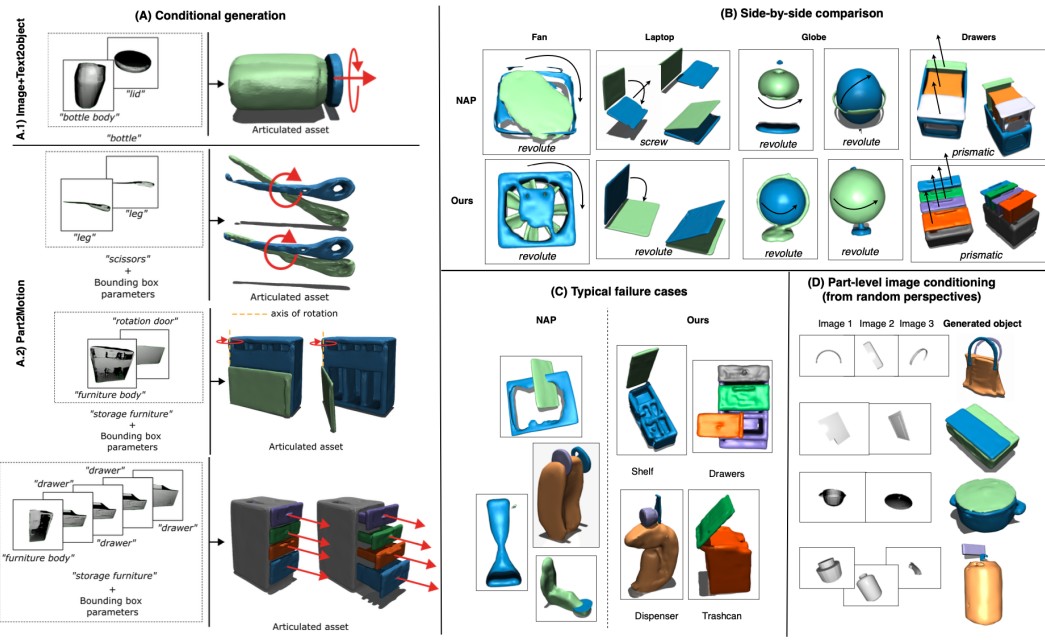

Figure 3: A) *PartToMotion* constrained structural generation. B) Side-by-side comparison between our approach and NAP. C) Typical failure cases. D) Part-level image conditioning.

Additionally, as shown in Figure 3.C, the absence of categorical information in NAP leads to the generation of objects composed of mismatched parts that do not cohesively fit together. In our approach, while there may be occasional issues with geometry or kinematics, the object parts remain semantically consistent, preserving the integrity of the overall structure. Finally, as illustrated in Figure 3.D, our conditional part-generation proves to be robust to images from different viewpoints, yielding enhanced flexibility and generalization capability of our method.

**Conditional Generation and Controllability** One of the key advantages of MIDGArD over existing approaches is its enhanced controllability, enabling users to guide the generation process using human-understandable graph attributes such as articulated asset types and body types. We showcase this capability in a "(Image+Text)-To-Object" setup (see Figure 3.A1), and a "Part-To-Motion" setup (see Figure 3.A2), where the model is provided with part features only (i.e., no joint data) and outputs consistent articulated assets. Notably, MIDGArD can also be controlled using only asset-level data; for instance, users can query for specific categories such as "storage furniture" or "fan" without needing to provide detailed information for each node (see Appendix C for additional qualitative results). Furthermore, the modular structure of our generation pipeline enables fine-grained, part-level control. Users can specify the desired appearance or attributes of individual parts by adjusting the human-interpretable articulation solutions synthesized by the structure generator accordingly. The resulting images and text description will then be used as conditioning signals for the shape generation module. As illustrated in Figure 4, varying the image inputs while keeping other attributes fixed results in generated parts that adapt accordingly, demonstrating the flexibility and responsiveness of our approach. This level of control allows for precise customization of generated assets, facilitating applications that require specific design features or aesthetic qualities (see Appendix D for additional qualitative results).

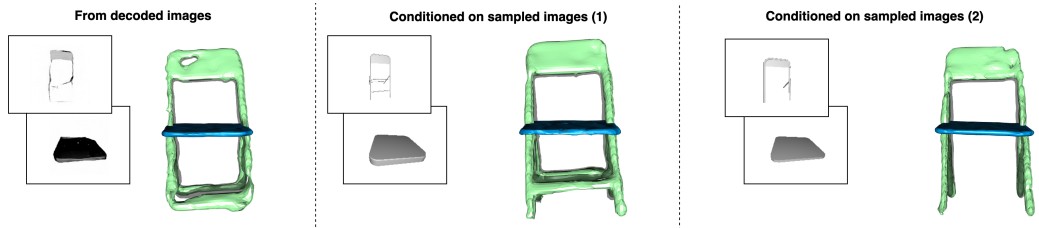

Figure 4: Image guidance of the shape generation process

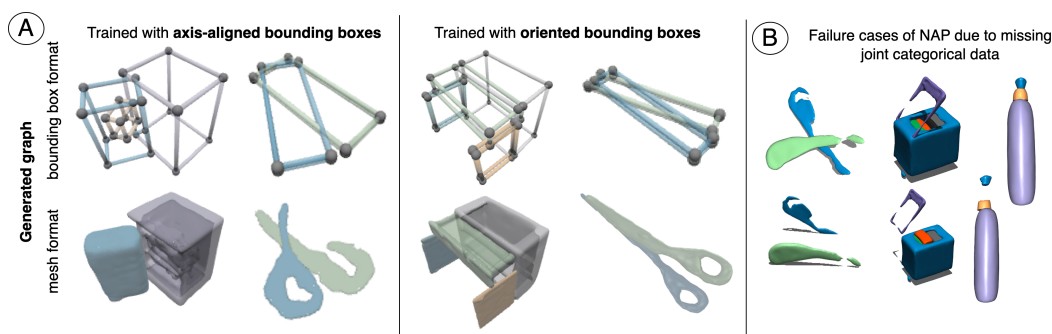

Figure 5: Failure cases due to due to axis-aligned bounding boxes and parameter discretization. A) The use of Axis Allligned Bounding Boxes tends to confuse the 3D generation model. B) The use of categorical joint variable in MIDGArD allows better joint motion resolution and helps filtering unrealistic solutions obtained with former pipelines.

**Constrained Shape Generation** We conducted an ablation study to evaluate the effectiveness of our bounding-box constrained shape generation approach introduced in section 3.3. The experiments were performed on *part*-samples from PartNet Mobility. We randomly sampled ten objects per category from the test data, including all their parts, resulting in 1044 samples across 32 categories. First, we measure whether the generated object fits within the bounding box. The model output $Y^{gen}$ is a sampled TSDF of resolution 64, $Y^{gen} \in \mathbb{R}^{64 \times 64 \times 64}$, where negative values indicate the object's interior. We compare $Y^{gen}$ with the TSDF of the bounding box $Y^{bb}$ and compute the count of all cells outside of the bounding box ($OBB_{count}$) and their sum of SDF values ($OBB_{sum}$):

$$OBB_{count}(O_{bb}, O_{gen}) = \sum_{ijk} \mathbb{1}[Y^{bb}_{ijk} > 0 \wedge Y^{gen}_{ijk} < 0], \tag{2}$$

$$OBB_{sum}(O_{bb}, O_{gen}) = \sum_{ijk} Y^{gen}_{ijk} \cdot \mathbb{1}[Y^{bb}_{ijk} > 0 \wedge Y^{gen}_{ijk} < 0]. \tag{3}$$

Secondly, we evaluate the generation quality after transforming the TSDF into a mesh, computing the Chamfer distance (CD) between generated and real objects based on 5000 sampled points. Table 3 compares our *bb prior* approach to the original SDFusion model trained on PartNet Mobility. For encoding the part dimensions, we use an MLP with three hidden layers (size 16, 64 and 256 respectively). For a comparison to a GAT-encoding, see Appendix E. Our first modification of preprocessing the dataset to rectify objects into canonical pose, dubbed "original + bb MLP + oriented dataset", already improves the generation results significantly, leading to lower $OBB_{count}, OBB_{sum}$ and CD. The *bb prior* method, however, consistently ensures accurate proportions and sizes for the generated links, diminishing erroneous volume. Additionally, *bb prior + bb MLP* method secures the best Chamfer Distance at 0.072. An alternative technique involves resizing generated objects to suit the bounding box dimensions. Such postprocessing of the model outputs reduces the average CD to 0.0061, close to matching that of the *bb prior* method. Nonetheless, such resizing may lead to significant distortions in object proportions, thereby underscoring the superiority of the *bb prior* method.

**Analysis of Oriented Bounding Boxes** As Table 3 shows, using oriented bounding boxes drastically improves the reconstruction performance. To quantify the change to the dataset, we measured the volume reduction of the bounding boxes after applying our PCA-based rectification approach. On average, the bounding box volume decreased by 17.4%, with one-third of the samples experiencing

Table 3: Ablation of bounding-box constrained shape generation. The original SDFusion pipeline, only enhanced with bounding-box conditioning, is compared to our approach. Training on a preprocessed dataset and post-processing the generated shapes afterwards already improves the performance significantly. The best results are achieved using our bounding-box SDF prior with post-processing.

|  | $OBB_{count}$ | $OBB_{sum}$ | CD (generated) | CD (scaled) |
|---|---|---|---|---|
| bb prior + bb MLP (Ours) | **1146** | **10.89** | **0.0072** | **0.0043** |
| original + bb MLP + oriented dataset | 1490 | 37.15 | 0.0152 | 0.0061 |
| original (SDFusion) + bb MLP | 3795 | 183.5 | 0.0307 | 0.008 |

a volume decrease of more than 10% and 20% of the samples experiencing a volume decrease of more than 30%. Figure 5 illustrates failure cases where using axis-aligned bounding boxes leads to unrealistic part shapes, such as a thick cabinet door.

## 5 Conclusions

In this work, we introduced MIDGArD, a novel framework for generating 3D articulated objects. MIDGArD employs a modular approach, combining interpretable articulation graph generation with high-quality shape generation. By leveraging categorical parametrization, MIDGArD enhances the consistency and plausibility of articulated objects, producing high-quality meshes with accurate dimensions through a constrained shape generation mechanism. The human-interpretable representation of images and text within the articulation graph allows for part-level and multi-modal control over the generation process. The models produced by MIDGArD are fully simulatable, paving the way for applications in text-guided or image-guided content creation.

MIDGArD addresses several issues identified in Neural 3D Articulation Prior (NAP) by improving joint conditioning with categorical embeddings, enforcing physical constraints via bounding-box constrained generation, and providing part-level control using latent image codes. Our results demonstrate MIDGArD's superior performance in terms of structural quality, consistency, and interpretability compared to existing methods. The framework's ability to generate diverse articulated objects based on partial graph inputs and shape conditioning underscores its potential for broad applications in digital content creation and robotics.

Despite the advancements introduced by MIDGArD, there are several limitations and areas for future improvement: 1) We observe that the current metrics for articulated generation are still very limited and benchmarks as well as evaluation frameworks must be developed for this field. For instance, Instantiation Distance is still based on point clouds, which ignores cases of *small* parts of the object with unrealstic motion, such as a cart wheel moving up to a meter away from the cart. 2) Overcoming limitations of graph scalability posed by node number constraints remains an important challenge. 3) Furthermore, it would be interesting to investigate the use of mixed integer noise methods such as MiDi [91] in the articulation graph generation pipeline to process the continuous and discrete variables. 4) Enhanced conditioning of the structure generation process using natural language or image data represents a highly promising avenue for future research. Foundational efforts in this area, such as those by Cai et al. [5], have laid the groundwork for further exploration. 5) Integrating specialized templates, such as human body poses, could significantly enhance MIDGArD's modelling versatility. Lastly, 6) testing alternative 3D generation models and adapting MIDGArD to handle different geometric primitives could broaden its generalization capabilities.

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

# A Details of the Training and Inference Process

## A.1 Training and Inference of the Structure Generator

As highlighted in [37, 46], the different characteristics of an articulated asset can be represented as features of a complete graph $\boldsymbol{G}_N = \{\boldsymbol{x}, \boldsymbol{e}\}_N$ with $N$ nodes, and $N(N-1)/2$ edges. We train an image VQVAE [89] using different views of every single object of the dataset. The latent code of every object $i$ composing an articulated asset is of dimension $8 \times 8$. This latent code is vectorized in an array $\boldsymbol{f}_i \in \mathbb{R}^{64}$ before being concatenated to the reference node feature vector $\boldsymbol{x}_i$ (see Figure 6).

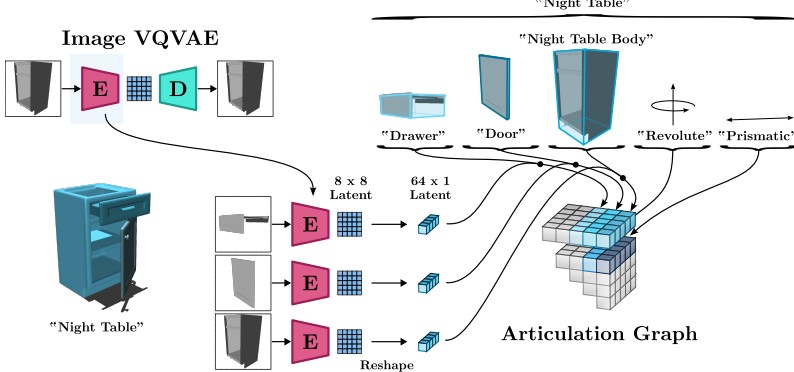

Figure 6: Detail of the structure generation training pipeline.

The node and edge features of the obtained asset graph $\boldsymbol{G}_N$ are then iteratively corrupted by Gaussian noise that a Graph ATtention network (GAT) learns to predict (see Figure 7).

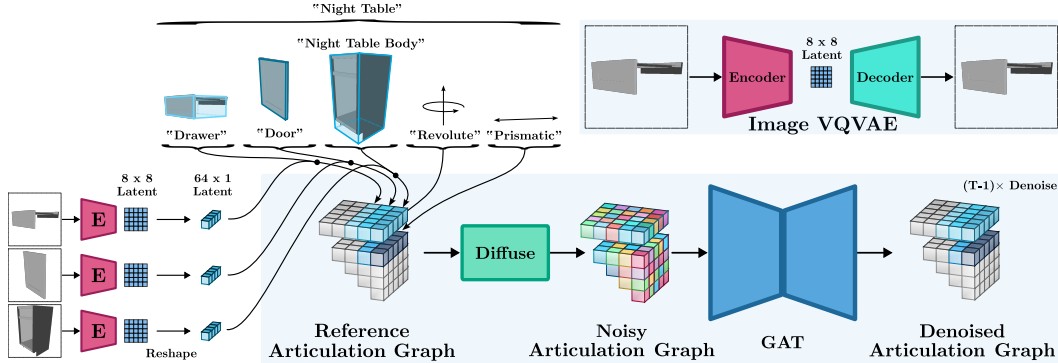

Figure 7: Detail of the structure generation training pipeline.

The denoised graph $\hat{\boldsymbol{G}}_N$ obtained at the output of the pipeline can then be post-processed into a *human interpretable* form, the node, joint and asset categorical information being converted into text while the denoised image latents $\boldsymbol{f}_i$ of every node $i$ are decoded into corresponding images. This human interpretable content can then be passed as a condition signal to the shape generator (see Figure 8).

## A.2 Denoising Diffusion Probabilistic Model (DDPM)

### A.2.1 Forward Process

Given a sample $\boldsymbol{x}_0$ drawn from a distribution $q\left(\boldsymbol{x}_0\right)$, a series of noisy samples $\boldsymbol{x}_t$, $\forall t \in [1, \cdots, T]$ can be obtained by gradually corrupting $\boldsymbol{x}_0$ with Gaussian noise $\boldsymbol{\varepsilon} \sim \mathcal{N}\left(\boldsymbol{0}, \boldsymbol{I}\right)$ following a variance schedule $\beta_1 < \cdots < \beta_T$:

$$q(\boldsymbol{x}_{1:T}|\boldsymbol{x}_0) := \prod_{t=1}^{T} q\left(\boldsymbol{x}_t|\boldsymbol{x}_{t-1}\right), \quad q\left(\boldsymbol{x}_t|\boldsymbol{x}_{t-1}\right) := \mathcal{N}\left(\sqrt{1-\beta_t}\boldsymbol{x}_{t-1}, \beta_t\boldsymbol{I}\right). \tag{4}$$

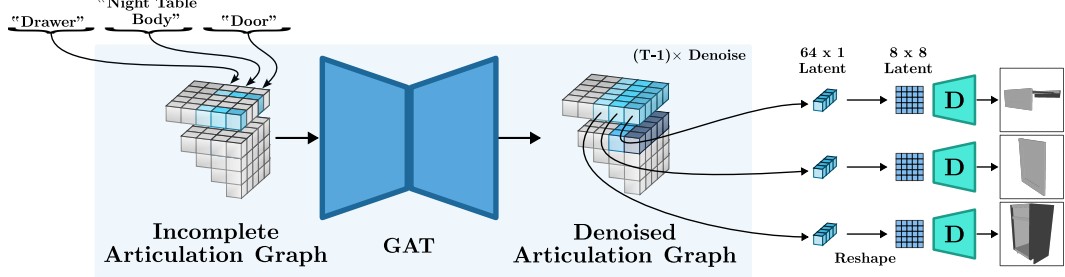

Figure 8: Detail of the structure generation inference pipeline.

A relevant property of such a diffusion process is that $\boldsymbol{x}_t$ can be sampled from $\boldsymbol{x}_0$ through

$$q(\boldsymbol{x}_t|\boldsymbol{x}_0) = \mathcal{N}(\sqrt{\bar{\alpha}_t}\boldsymbol{x}_0, (1-\bar{\alpha}_t)\boldsymbol{I}), \tag{5}$$

where $\alpha_t := 1 - \beta_t$ and $\bar{\alpha}_t = \prod_{s=1}^{t} \alpha_s$, eventually yielding $\boldsymbol{x}_t = \sqrt{\bar{\alpha}_t}\boldsymbol{x}_0 + \sqrt{1-\bar{\alpha}_t}\varepsilon$.

### A.2.2   Reverse Process.

Initiating from a standard Gaussian distribution, $\boldsymbol{x}_T \sim \mathcal{N}(0, \boldsymbol{I})$, a denoising model $p_\theta$ parameterized by trainable weights $\theta$, learns to approximate a series of Gaussian transitions $p_\theta(\boldsymbol{x}_{t-1}|\boldsymbol{x}_t)$. These transitions incrementally denoise the signal such that

$$p_\theta(\boldsymbol{x}_{0:T}) \quad := \quad p_\theta(\boldsymbol{x}_T)\prod_{t=1}^{T} p_\theta(\boldsymbol{x}_{t-1}|\boldsymbol{x}_t), \tag{6}$$

$$p_\theta(\boldsymbol{x}_{t-1}|\boldsymbol{x}_t) \quad := \quad \mathcal{N}(\boldsymbol{\mu}_\theta(\boldsymbol{x}_t, t), \boldsymbol{\Sigma}_\theta(\boldsymbol{x}_t, t)). \tag{7}$$

Following the approach from [21], previous work on articulated asset generation [37] [46] define $\boldsymbol{\mu}_\theta = \frac{1}{\sqrt{\alpha_t}}\left(\boldsymbol{x}_t - \frac{\beta_t}{\sqrt{1-\bar{\alpha}_t}}\varepsilon_\theta(\boldsymbol{x}_t, t)\right)$ and $\boldsymbol{\Sigma}_\theta(\boldsymbol{x}_t, t) = \sigma_t^2\boldsymbol{I}$ yielding the following Langevin dynamics

$$\boldsymbol{x}_{t-1} = \frac{1}{\sqrt{\alpha_t}}\left(\boldsymbol{x}_t - \frac{\beta_t}{\sqrt{1-\bar{\alpha}_t}}\varepsilon_\theta(\boldsymbol{x}_t, t)\right) + \sigma_t\boldsymbol{z}, \quad \boldsymbol{z} \sim \mathcal{N}(\boldsymbol{0}, \boldsymbol{I}), \tag{8}$$

where $\varepsilon_\theta(\boldsymbol{x}_t, t)$ is a learnable network approximating the per-step noise on $\boldsymbol{x}_t$.

### A.2.3   Loss Function

The variational lower bound is used to optimize the negative log-likelihood. Following the simplified training objective outlined in DDPM [21], the training loss can be simplified to the following expression

$$\mathcal{L} = \mathbb{E}_{\boldsymbol{x}_0, \varepsilon_t}\left[\left\|\varepsilon_t - \varepsilon_\theta\left(\sqrt{\bar{\alpha}_t}\boldsymbol{x}_0 + \sqrt{1-\bar{\alpha}_t}\varepsilon_t, t\right)\right\|^2\right]. \tag{9}$$

# B  Data Preprocessing

## B.1  Mesh Preprocessing

High-quality meshes are essential for applications such as generating Truncated Signed Distance Functions (TSDFs), which are in turn widely used within advanced 3D latent diffusion models such as SDFusion [8]. However, many meshes in popular datasets, including PartNet-Mobility [97], contain inconsistencies and errors that render them unsuitable for such purposes. To address these issues, we incorporate a preprocessing routine that includes a manifoldization process [26] to repair and convert flawed meshes into watertight, manifold counterparts (see Figure 9).

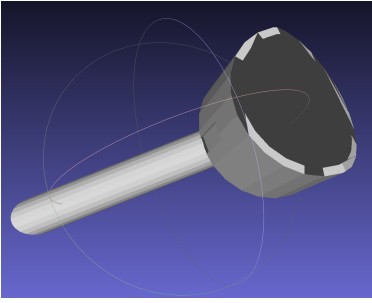 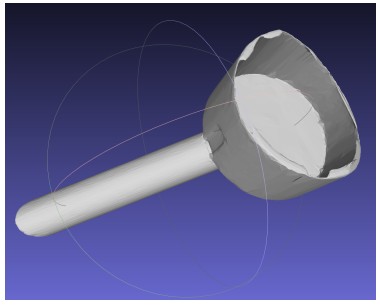

(a) Original mesh with inconsistencies          (b) Repaired watertight manifold mesh

Figure 9: Mesh repair pipeline: (a) The original mesh exhibits holes and non-manifold edges; (b) After preprocessing, the mesh is converted into a watertight manifold suitable for TSDF generation.

## B.2  Orientation Estimation and OBB Computation

Unlike previous approaches [37, 46], we do not assume that the dataset meshes are already provided in a canonical orientation. Our method hence involves an orientation estimation process for each individual mesh, performed through the calculation of its Oriented Bounding Box (OBB). This OBB is subsequently utilized to align each body into a standard pose. The orientation estimation begins by

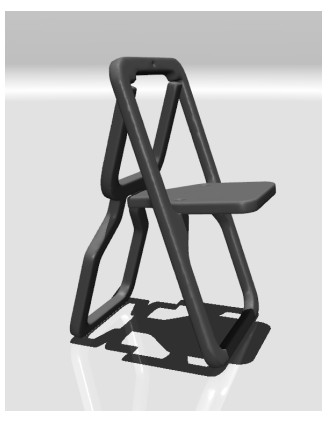 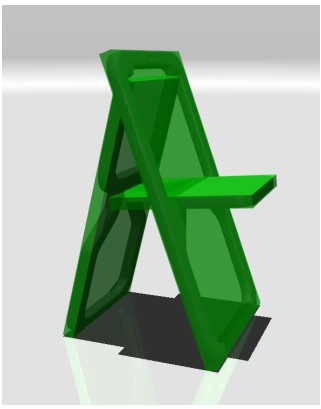 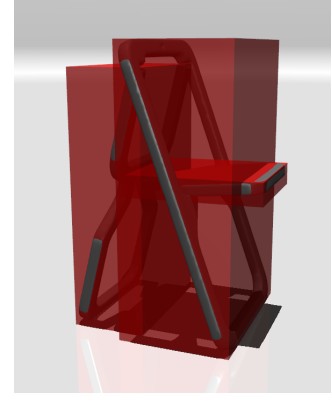

(a) Original mesh          (b) Mesh with computed OBBs          (c) Mesh with original AABBs

Figure 10: Comparison of bounding boxes: (a) The original mesh; (b) The mesh enclosed by Oriented Bounding Boxes (OBBs), providing a tight and accurate fit; (c) The mesh enclosed by original Axis-Aligned Bounding Boxes (AABBs).

applying Principal Component Analysis (PCA) to the mesh data. To mitigate the influence of complex internal geometries, such as shelves within furniture models, we first compute the convex hull of the mesh. This step simplifies the geometry while preserving the overall shape. We then voxelize the convex hull and uniformly sample 10,000 volumetric points from it. Applying PCA to these sampled points provides an initial estimate of the mesh's principal axes and orientation. Building upon this

initial estimate, we refine the attitude quaternion of the OBB through gradient descent optimization. The objective is to minimize the volume of the OBB, achieving a tighter fit around the mesh. During optimization, we iteratively adjust the orientation by recalculating the OBB volume along the oriented principal axes. To reliably navigate the solution space and avoid local minima, we incorporate a heuristic that explores potential orientations within a cone defined by an angle $\theta$ centered around the current best estimate. This approach allows for nuanced adjustments and aids in converging to a global minimum, thereby enhancing the precision of the orientation estimation. By combining PCA for initial alignment with heuristic-augmented gradient descent for volume minimization, our method ensures accurate and robust alignment of dataset meshes into their canonical poses (see Figure 10). This preprocessing step significantly enhances the overall quality and consistency of the articulated objects generated by our framework.

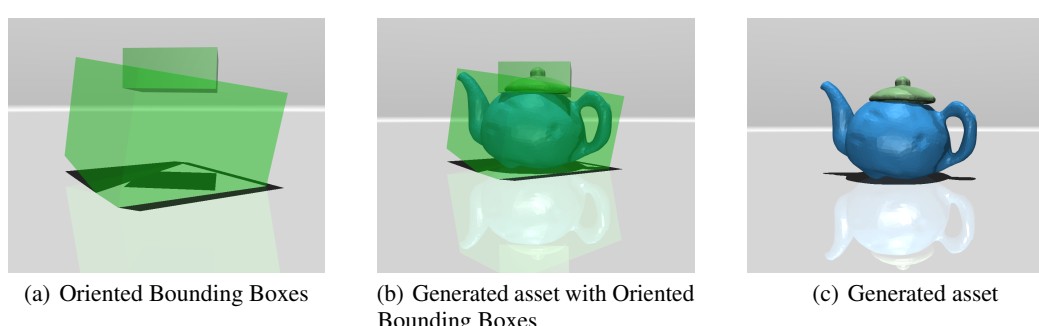

(a) Oriented Bounding Boxes    (b) Generated asset with Oriented Bounding Boxes    (c) Generated asset

Figure 11: An articulated asset simulated in the MuJoCo[86] physics engine.

## C    Controlling Generation with Asset Labels

MIDGArD incorporates categorical embeddings for asset types and object categories directly into the node features of the articulation graph. These embeddings enable the model to understand and generate objects that belong to a specified category, capturing the common structural and kinematic features associated with that category. The model is trained on a diverse dataset that includes various object categories with different articulation patterns. By learning statistical priors over these categories, MIDGArD can infer plausible structures and motions for new objects within the same category. The structure generator is designed to handle incomplete or partial inputs, allowing it to fill in missing details based on learned patterns. In the absence of detailed node-level information, the model can perform unconditional generation, producing coherent articulation graphs solely based on the provided asset category. The use of categorical embeddings and latent codes that correspond to human-understandable concepts (like "storage furniture") makes it easier for users to interact with and guide the generation process.

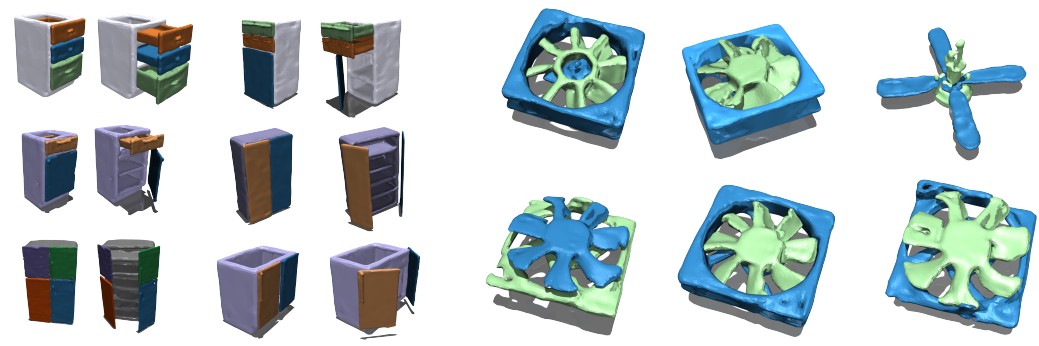

(a) Controlled generations with asset label "storage furniture"    (b) Controlled generations with asset label "fan"

Figure 12: Controlled Generation with Asset Labels.

# D    Controlling Generation with Image Input

A key advantage of MIDGArD's modular setup is the controllability of the generative process, not only on a graph level (by supplying partial graphs) but also on a part-level, by replacing the generated image or text information. In Figure 13, we provide examples how modifying the input image to the shape generator (SDFusion) changes the object appearance. Here, we randomly choose images from PartNet Mobility, sampling among all parts with the same category. For example, we sample from from all meshes with the descriptions "*A wheel as part of a cart*" to generate the wheel in the fifth row of Figure 13.

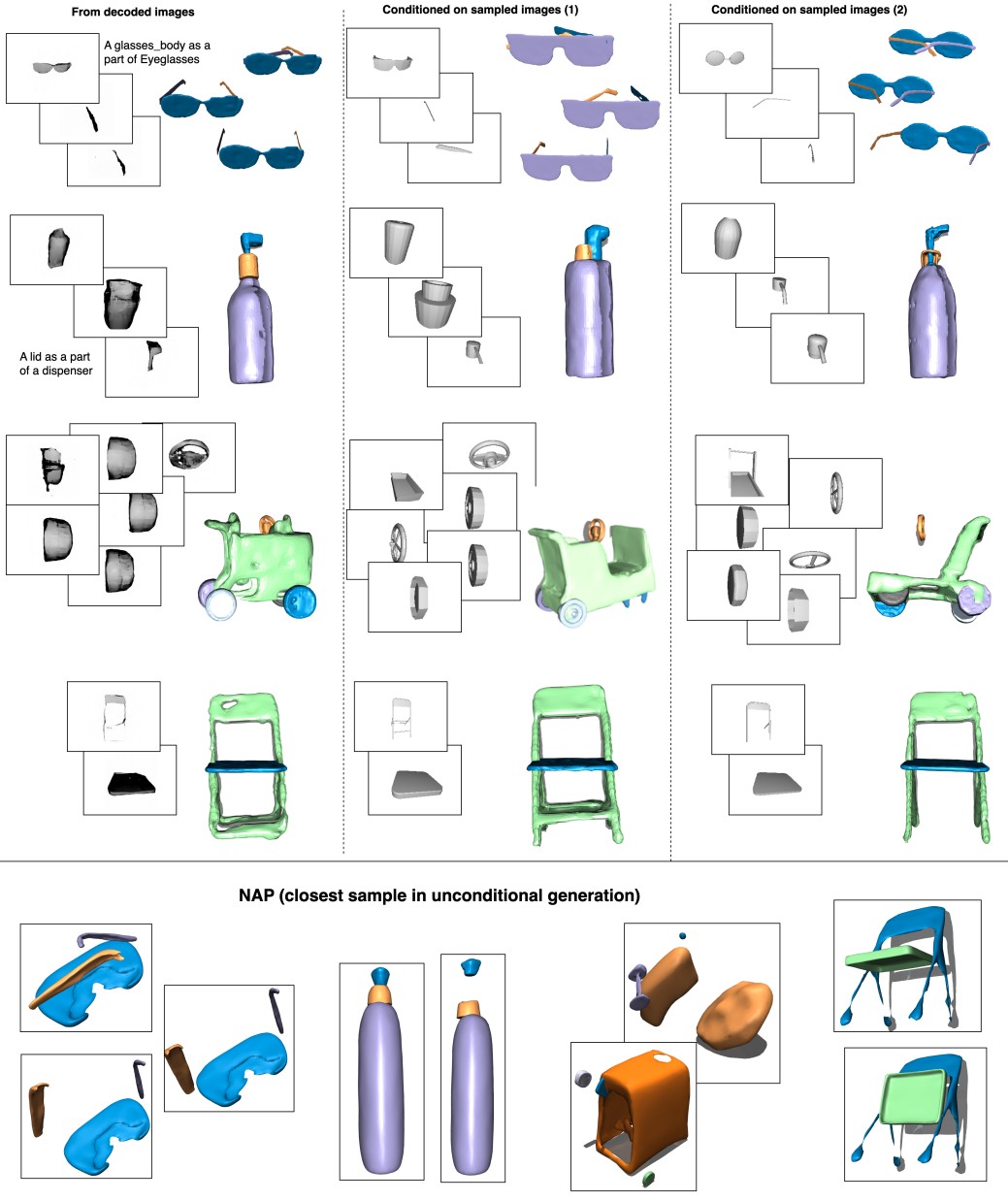

Figure 13: Controlling the shape appearance of object parts via image conditioning. Top: Image inputs allow to control the object design on a part-level. Bottom: NAP struggles to generate objects with many parts and yields unrealistic motion.

# E   Encoding object-level information with Graph Attention Networks

For capturing the relationships between parts, we experiment with graph encodings as input to the conditional shape generation process. To generate the geometry for a specific part $\phi$, we construct a simplified graph from the articulation graph that provides information about the size of the $\phi$ in comparison to other parts, as well as the joints between $\phi$ and other parts. Specifically, let $G_\phi = (\boldsymbol{v}_\phi, \boldsymbol{e}_\phi)$ be a graph with $t_i$ as the node feature vector for the i-th node and the Plücker coordinates $p_{ij}$ serving as edge features. We test two options to mark $\phi$ as the part to be generated: (1) a tree structure where $\boldsymbol{v}_{phi}$ is constructed by reducing the graph to the parts adjacent to $\phi$, and (2) a complete graph ($\boldsymbol{v}_\phi = \boldsymbol{v}$) where the node features are augmented with a binary indicator that equals 1 for the node representing $\phi$. These structures are processed with a Graph Attention Network (GAT), which is trained concurrently with the diffusion model. As shown in Table 4, the results do not match the ones of a simple MLP embedding, neither with nor without our *bb prior* method. We hypothesize that this is due to concurrently training the GAT with the diffusion model, possibly converging slower than an MLP. Furthermore, many objects in PartNet Mobility have few parts, leading to small graphs of less than 4 nodes. Future work could try to pretrain a GAT on graph-encoding to improve the conditional generation with object-level information.

Table 4: Shape generation results contrasting an MLP-based embedding to a GAT encoder.

|  | $OBB_{count}$ | $OBB_{sum}$ | CD (generated) | CD (scaled) |
|---|---|---|---|---|
| bb prior + bb MLP | 1146 | 10.89 | 0.0072 | 0.0043 |
| bb prior + tree GAT | 1246 | 10.97 | 0.0076 | 0.0044 |
| bb prior + graph GAT | 1226 | 13.12 | 0.0084 | 0.0046 |
| original + bb MLP + oriented dataset | 1490 | 37.15 | 0.0152 | 0.0061 |
| original + oriented dataset + tree GAT | 1994 | 69.66 | 0.0203 | 0.0051 |
| original + oriented dataset + graph GAT | 1710 | 49.42 | 0.0218 | 0.0052 |
| original + bb MLP | 3795 | 183.5 | 0.0307 | 0.008 |

