# OpenReview forum: "MIDGArD: Modular Interpretable Diffusion over Graphs for Articulated Designs"
_NeurIPS.cc/2024/Conference — NeurIPS 2024 poster_

### Official Review · Reviewer_czmP · 2024-06-23

**Soundness:** 3
**Presentation:** 2
**Contribution:** 2
**Rating:** 5
**Confidence:** 4

**Summary:**

This paper presents a diffusion-based framework for generating articulated objects represented as part graphs. Different from prior work [35] that generates both the part shapes and graph structures simultaneously, this work introduces a two-stage strategy that first generates the structure with per-part conditions and then generates the part shapes. It also proposes a part latent encoding that enables multimodal conditions including text and images.

**Strengths:**

- The reparameterization of the Plucker coordinates is simple but more natural than that in [35], making the diffusion results better satisfying the constraints.
- The designed node embedding with image and text features is not only a more detailed representation of part shapes, but also enables more flexible conditioned generation for different downstream applications.
- The framework relaxes the constraints of training on canonical-posed shapes. This is an important relaxable for better usage of data.

**Weaknesses:**

- As also briefly discussed in the paper, generating the parts independently is a bit unnatural to this problem, especially given the importance of inter-part motions/relations in articulated objects. But overall I am OK with this point given all the node conditions in the framework.
- Quantitatively, the improvements compared to [35] on unconditioned generation seem marginal.
- The description "simulation-ready" sounds overclaimed to me. When talking about "simulation", I feel most people will expect physical viability and accuracy. But from the demonstration on the website, it seems that the parts are not well connected. There is also no explicit physics enforcement in the framework.

Minor things:
- I believe it would be better to swap the two sentences "This image latent is derived by" (line 148) and "...the structure generator denoises a latent representation of an image..." (line 150), so the readers know where the "image latent" in line 148 comes from.

**Questions:**

I am curious about some details:
- Sec. 3.4 says "During training, the model uses renderings of the part mesh from a frontal view" (line 217). How is the frontal view chosen? For inference-time image-conditioned generation, do the images have to be in frontal view, or can they be of any view? Are the images in Suppl. Fig. 5 of "frontal view"?
- I wonder how exactly are the bounding boxes represented and predicted. Because for articulated objects, the part poses are changing during part motions. [35] only uses the rest-state bounding boxes (together with the joint limits). In this work, is the rest-state bounding box viewed as having the identity pose matrix?

**Limitations:**

Limitations are well-discussed in the paper.

---

> ### Author Rebuttal · Authors · 2024-08-07
>
> **Part-level generation**
>
> Part-level generation requires conditioning on the articulation graph, since the appearance of a part depends on its role within the graph. In future work, we aim to test further possibilities to condition the part-generation on already existing parts, e.g. by framing it as a shape completion problem. However, we chose this approach since it also has several benefits:
>
> * As already noted by the reviewer, it allows for part-level control via text or images.
>
> * learning to generate object parts is arguably simpler than learning to generate arbitrary objects. Generating a complex high-resolution object is still an open challenge in 3D generation, but generating basic components of objects such as wheels and boxes is feasible.
>
> **Unconditional generation**
>
> NAP is already a rather compact and powerful model but the quality measurement metric (Instantiation Distance or ID) has limitations, as it mixes shape geometry with structure and motion evaluations; two properties that should be separately measured. Since points are sampled randomly for the CD, this ignores unrealistic states of small parts, such as the lid of the bottle in Fig 4. This is an issue that was also raised in CAGE. However, for a fair comparison to NAP's results, we used their evaluation protocol. We provide a qualitative comparison between NAP and MIDGArD in the attached PDF, Figure (B). We also provide further quantitative results to show the improvement of our approach in terms of physical plausibility, we compute the distribution of joint types and compare the distribution among our generated objects with the ones from NAP and the real data. As shown in the main rebuttal section, our model learns to match the distribution well. Since we determine the joint ranges based on the type, this directly leads to an improved motion plausibility. Finally, it is worth noting that one of our contribution is the higher level of control achieved with our approach, offering novel ways to guide the generation of articulated objects. Please refer to the attached PDF, specifically Figure (A) in the main rebuttal section.
>
> **Simulation ready sounds overclaimed.**
>
> By "simulation-ready", we refer to our contribution of providing the possibility to export the result to Mujoco, which is indeed a physically accurate simulation environment. Building a MuJoCo environment with multiple articulated assets will be straightforward with our codebase, which is why “simulation-ready” seemed to be a suitable term. Nevertheless, we do not want to neglect the huge effort involved with building full simulations, so we will weaken “simulation-ready” to “simulatable” in our revised manuscript.
>
> **It seems that the parts are not well connected**
>
> We believe the reviewer refers to the fact that screws and other connectors are not shown in the simulated videos. The connectors are omitted since they are not part of the ground truth dataset. However, adding those would be straightforward based on the predicted edge features in the graph (joint type \& Plucker coordinates).  The crucial part of the simulation in our point of view is the motion of the object parts, which is realistic in our videos. Please let us know if we have addressed your concerns or if any further clarifications would be useful.
>
> **There is no explicit physics enforcement.**
>
> We agree that there is no strict enforcement of physics in the framework; however, this is a rather difficult endeavor as it usually requires the interaction of the learning pipeline with a physics engine. Meanwhile, several components of our framework ensure physical plausibility:
>
> * Our bounding-box constrained generation approach (with subsequent part scaling and orientation) ensures that the parts fit within the object in a physically plausible way.
>
> * We enforce realistic kinematics by specifying the joint ranges based on the joint category. As Fig 4B shows, this works much better than the approach taken in NAP, i.e., predicting the joint range directly.
>
> **Frontal-view images**
>
> While we use a specific perspective for training and testing in the initial submission, we provide additional results here showing that our model generalizes to image inputs from other perspectives (see Fig. D in the main rebuttal section).
> The PartnetMobility dataset is already normalized such that all parts are shown from the front (if such asymmetry even exists). However, rendering with a camera angle of [0,0,0] is problematic since some objects are not recognizable from the front; e.g., a wheel may appear as a rectangle from the front. Therefore, our “frontal view” refers to a rendering from an axis angle [\phi/6,  \pi / 12, 0] which usually shows the relevant part geometry. We've added this clarification in the manuscript.
> Due to the variety of parts in our training dataset, we hypothesize that our model should also generalize to images from other views. We tested this by conditioning the model on images from a randomly sampled perspective.
>
> **Are the images in Suppl. Fig. 5 of "frontal view"?**
>
> Yes, the images in Suppl Fig 5 are rendered with the same angle as the images used for training
>
> **Bounding boxes**
>
> In the articulation graph, the part-bounding box is represented by three parts: 1) its center, 2) its size 3) its rotation. These three components are generated for each part in the structure generator. The shape generator, in turn, is conditioned on the desired size (2) and generates a centered object of the correct size. The rotation (3) is applied afterwards. Note that this approach allows to generate parts in arbitrary orientation, in contrast to previous approaches (CAGE), that assumes that all parts have axis-aligned bounding boxes when the full object is in resting state.

---

> > ### Comment · Reviewer_czmP · 2024-08-13
> >
> > Thanks for the detailed responses. My concern about the evaluation and baseline comparisons is very well addressed. I will keep my positive rating. I would also encourage the authors to add these explanations (especially about the "frontal view") to the paper in their revisions.

---

> > > ### Author Response · Authors · 2024-08-14
> > >
> > > Thank you very much for your kind words and positive rating.
> > > We are glad to hear that your concerns about the evaluation and baseline comparisons have been well addressed.
> > > We will integrate this feedback to improve the clarity and comprehensiveness of our work.

---

### Official Review · Reviewer_2qcf · 2024-07-11

**Soundness:** 3
**Presentation:** 3
**Contribution:** 4
**Rating:** 7
**Confidence:** 3

**Summary:**

This work tackles the problem of generating articulated 3D assets that are animatible. The authors mention that their generated shapes are directly compatible with existing physics simulation tools, i.e. MuJoCo. To this end, they first propose a structure generator, which conditionally or unconditionally generates an articulation graph encoding structural object information such as the kinematics attributes, i.e. the kinematic tree and joint types. Next, the multi-modal shape generator synthesizes a shape that follows the kinematic tree and size of the object while also allowing conditioning on multi-modal inputs such as text and images. Results demonstrate some improvement over the SOTA (NAP).

Overall, I like the setting that the authors propose as well as their technical contribution to solve it. There are a few minor concerns about the presentation in terms of writing, clarity, and qualitative visualizations (see comments above). However, I believe those are minor. Thus, I recommend acceptance.

**Strengths:**

- Interesting setting with a lot of potential for future work. It seems only very limited amount of works (NAP) have explored this setting
- Evaluation is to the best of my knowledge complete.
- Well-written related work section and references seem to be complete
- The paper is technically sound

**Weaknesses:**

- Writing:
    - I would limit the contribution bullets to the technical contribution of this work rather than focusing on results and open sourcing.
- Clarity
    - I would introduce proper notations for all steps discussed in section 3.3 as I feel it is hard to understand the setting just from the textual descriptions, e.g. what is input, what is output.
    - I have a similar concern regarding section 3.4
- Video
    - I would have expected to also see some video results of the generated and animated objects. While this is not strictly required, it would have made the exposition much more complete.

**Questions:**

--

**Limitations:**

Limitations are sufficiently discussed in the main paper.

---

> ### Author Rebuttal · Authors · 2024-08-07
>
> Thank you for the constructive feedback that help us to improve the clarity and quality of our paper.
>
> **Writing (comment: I would limit the contribution bullets to the technical contribution of this work rather than focusing on results and open sourcing)**:
>
> Thank you for your suggestion. We fully agree and will replace bullet points 4-6 with the following technical contributions:
> * An approach for constrained shape generation within oriented bounding boxes that improves alignment of the kinematic links by 17\% on average.
> * A pipeline to create fully simulatable assets with an interface to Mujoco
>
> **Notation and clarity (comment: I would introduce proper notations for all steps discussed in section 3.3 as I feel it is hard to understand the setting just from the textual descriptions, e.g. what is input, what is output. I have a similar concern regarding section 3.4)**
>
> Thank you for this suggestion to improve the clarity of the paper. We will rewrite this part to improve the description of the setting. Specifically, we will add that we train the model to learn the distribution of articulated object graphs, by applying a denoising diffusion process:
> * Input: Noisy Graph $\boldsymbol{\tilde{G}}_N = \left\lbrace \boldsymbol{\tilde{x}}, \boldsymbol{\tilde{e}} \right\rbrace$ with node and edge features as introduced in section 3.3. For conditional generation, the input is a partially noisy graph, where certain node or edge features are masked.
> * Output: Denoised articulation graph $\boldsymbol{G}_N = \left\lbrace \boldsymbol{x}, \boldsymbol{e}\right\rbrace_N$
>
> Similarly, for section 3.4, we will add the following description:
>
> The aim of the structure generator is to learn the conditional probability distribution $P(z_i | a_i, b_i, d_i, r_i, t_i, g_i)$ for generating the latent representation $z_i $ of a part geometry's Signed Distance Function (SDF) using a diffusion model, conditioned on inputs $[a_i, b_i, d_i, r_i, t_i, g_i]$ as explained in section 3.3.
>
> This is achieved 1) by transforming $a_i$ and $b_i$ into a text description and encoding with the BERT model, 2) by transforming the image (decoded $g_i$) via a ResNet and 3) by applying constrained generation with $r_i$ and $t_i$ as explained in section 3.4
>
> **Video (comment: I would have expected to also see some video results of the generated and animated objects. While this is not strictly required, it would have made the exposition much more complete.)** :
>
> We have provided videos in a repository here: https://anonymous.4open.science/r/MIDGArD-E1DE/README.md (see folder “gifs”). We apologize if the reference to this repository was unclear in the paper. Upon publication, we will provide a proper website with these examples in addition to the open-source code.

---

> > ### Comment · Reviewer_2qcf · 2024-08-09
> >
> > Thank you for your detailed response.
> >
> > After reading the rebuttal and other reviews, I am still convinced that this work is suitable for Neurips.
> >
> > Therefore, I keep my original rating.

---

> > > ### Author Response · Authors · 2024-08-13
> > >
> > > Thank you for your detailed response and positive evaluation; your support for our work's suitability for NeurIPS is greatly appreciated.

---

### Official Review · Reviewer_jtcL · 2024-07-12

**Soundness:** 3
**Presentation:** 3
**Contribution:** 3
**Rating:** 7
**Confidence:** 4

**Summary:**

This paper proposes several interesting improvements over existing articulated object modeling and highlights higher-quality part generation.

Specifically, an articulated object is parameterized to a graph where parts are nodes and joints are edges. This paper proposes a multi-model part VAE for generating higher quality and controllable shapes and improving physical awareness through a bounding box alignment. It also improves the kinematic structure generation by considering parameterize on the plucker manifold. With these technical contributions, the results show significant improvement over baselines and the supplemental repo provides simulatable mujoco files, showing the practical value of this work.

I recommend a clear acceptance of this paper.

**Strengths:**

- Good results, supp MuJoCo simulation: I really like the MuJoCo demo in the suppl repo, which is impressive.
- Several technical solid and important improvements including the shape representation, plucker manifold encoding, and bounding box alignment. etc
- Multimodel information and condition: i specifically like the text and language part of the part shape encoding, which opens a lot of opportunities for conditioned generation and connection to LLM.

**Weaknesses:**

- More conditioned generation results: while the part-level multimodel information is used in shape encoding, it would be better to show more conditioned generation results using this information.
- Open source the part text and bounding box alignment data: it's not clear whether the shape part encoding training data will be publicly released or not.

**Questions:**

Please see the weakness

**Limitations:**

Adequately discussed in the paper.

---

> ### Author Rebuttal · Authors · 2024-08-07
>
> Thank you for highlightning the strength of our approach. We are delighted that you find the MuJoCo simulations in the supplementary repository impressive and recognize the significance of our technical improvements, including shape representation, Plücker manifold encoding, and bounding box alignment. These were key motivations in our effort to advance the field of articulated object modeling.
>
> **Conditioned generation results: while the part-level multimodal information is used in shape encoding, it would be better to show more conditioned generation results using this information.**
>
> We acknowledge your request and have included additional conditioned generation results in the attached PDF, specifically Figure A ("Conditional generation"). This figure showcases our model's capability to generate consistent articulated assets based on supplied part features in form of image and text. In addition to the experiment in Figure 5 of the original submission, this figure shows two settings: Generating an articulated object solely based on image and text input, and generating the articulated object based on image, text and bounding box input. The latter is comparable to the "Part2Motion" setup in NAP. However, in NAP one would have to provide the full geometry for each part, whereas we demonstrate the same capability just based on human-interpretable input features.
>
> Furthermore, we have extended our experiment on image-based conditioning (Figure 3 and 5 of the original manuscript) by another experiment using images from arbitrary viewpoints for conditioning (Figure D in the attached PDF). This shows the generalisation capability of our approach.
>
> We hope these experiments match the reviewer's expectations and we welcome further suggestions to improve our empirical results.
>
> **Open source: it's not clear whether the shape part encoding training data will be publicly released or not.**
>
> Thank you for your suggestions. We commit to open-source the data (text, bounding box etc) and source code upon the paper's acceptance facilitating reproducibility and further research in this area. This commitment aligns with our goal of promoting transparency and accessibility within the research community.

---

> > ### Comment · Reviewer_jtcL · 2024-08-07
> > **Keep my original positive score**
> >
> > After reading the reviews and author response, the reviewer feel very good that the area of articulated object modeling is blooming. I believe this paper provides practically much more improvement over NAP. I keep my original score.

---

> > > ### Author Response · Authors · 2024-08-13
> > >
> > > Thank you for your encouraging feedback.

---

### Official Review · Reviewer_T3Uj · 2024-07-16

**Soundness:** 1
**Presentation:** 2
**Contribution:** 1
**Rating:** 3
**Confidence:** 5

**Summary:**

This work addresses the task of 3D asset generation for articulated objects.

This work aims to enhance the prior approach by achieving three main objectives: 1) increasing the interpretability and controllability of the generation process; 2) generating more natural joint motions and 3) reducing violations of physical constraints.

For the first goal, they propose a framework MIDGArD to decompose the generation into two sequential modules: 1) a structure generator that unconditionally generates a kinematic graph and attributes for each articulated part; 2) a shape generator to produce part SDF that allows conditioning on the graph from the first module, an image, and a text description for each part.

For the second goal, they introduce a representation for joint parameters that allows the diffusion process to operate on the Plucker manifold directly.

For the third goal, they replace the AABBs with OBBs to bound the part shape which is claimed to achieve better part alignment.

**Strengths:**

- This work contributes to an increasingly important area and identifies three valuable perspectives for articulated object generation.
- The proposed framework that separates the generation of the articulation structure and part geometry by using images as an intermediate proxy allows users to control the part shape more explicitly, compared with the prior work NAP [35] which requires modifying the latent code instead.
- Both quantitative and qualitative evaluations are provided to demonstrate improved quality of data distribution modeling in the unconditional generation setting.

**Weaknesses:**

- The technical contribution is limited:
	- The network architecture of the structure generator is the same as NAP [35]. The contribution for this part is only the representation alternation for joint parameters and bounding boxes. These are reasonable changes for slightly better distribution modeling, but the insight is not particularly significant.
	- The shape generator is also adapted from an existing work SDFusion [7], where the benefits of detailed geometry modeling with TSDF and the flexibility of multimodal conditioning are just inherited from the original work. The only modification is an additional conditioning from a graph. However, how this graph condition affects the part geometry output and whether it is necessary is unknown. Also, there is no strong/sufficient qualitative evidence to show a better geometry generation compared to NAP.
- The experiments and evaluation are insufficient and unclear:
	- The side-by-side qualitative comparison with prior work (e.g. NAP [35], CAGE [43]) is missing to convince the improvement in the claimed aspects.
	- Unfair comparison with NAP in the “reconstruction” setting in Table 1. According to the description in lines 280-282, NAP is only provided with graph topology and motion attributes for each node. In this case, there is no way for NAP to reconstruct the object with no shape information (which should have been given the latent code extracted from the ground truth part geometry). In contrast, MIDGArD is additionally provided with shape features extracted from the ground truth node images.
	- Lack of quantitative and qualitative results to support the argument of more “natural joint motion” and “less physical constraint violation” and how these improvements are correlated with the specific designs. Other ablation studies discussed in the manuscript are also generally hard to follow.
- The related work section does not contextualize this work very well.

**Questions:**

- The argument in lines 47-48 “leveraging categorical embeddings for improving consistency across the graph” is unclear. What does consistency mean exactly? The ablation experiment to support this argument is missing.
- Would it be possible to compare with CAGE [43] on certain aspects of the generation? E.g., in terms of the physical plausibility of objects in both abstraction level (bounding boxes) and mesh output.
- The argument in line 33 “both NAP [35] and CAGE [43] provide no control on the part level due to their opaque encoding as a latent” is a bit misleading. What specific control on part is missing from the prior work? Also, to correct the fact, the representation used in CAGE has no latent encoding for any attribute, while the representation used in NAP only makes the geometric feature encoded as latent and leaves others in an explicit form.
- How robust and generalizable the image control is?

**Limitations:**

- Under the formulation of this work, the image control from users is not particularly practical:
	- The image of a single articulated part is not natural to find easily.
	- The control is more of a post-editing. The user has to wait to see what object is generated and then find compatible part images to feed into the corresponding node in the graph. Otherwise, it won’t be matched with the articulation parameters and the part arrangement being generated.

---

> ### Author Rebuttal · Authors · 2024-08-07
>
> **Weakness 1.1** While our structure generator builds up on NAP \cite{lei2023nap} and also applies a denoising diffusion process on the graph, the representation and generative process are fundamentally different:
> 1.) Our approach *modularizes* articulated asset generation. This fundamentally diverges from the approach taken in NAP which aims to learn both shape and structure generation within one model. While both the shape generator and the structure generator build up on previous work, they are substantially modified to enable their interplay for generating articulated assets. Only this modular approach enables fine-grained control on the object level (Table 1 and Fig. A in the attached PDF) and part level (Fig. 3, Fig. 5 appendix). 2.) The graph representation is different to NAP's representation. In the node features, the geometry encoding is removed, the image encoding and categorical variables are added, and bounding boxes are replaced by oriented bounding boxes. Only the part-existence indicator is the same. For edge features, joint category is added and we directly denoise on the Plücker manifold. 3.) These changes enable more control and enhances intuitiveness for the use (see examples for conditional generation, Figure A in the attached PDF).
>
> **Weakness 1.2** We respectfully disagree. As outlined in Section 3.4, we do not only supply the graph as another conditioning input, but modify the SDFusion pipeline to generate only the *difference* to the bounding box (see Figure 2). Thus, the core innovation within the shape generation pipeline is the introduction of a bounding box constraint. To the best of our knowledge, this is novel. This constraint, combined with the use of oriented bounding boxes (OBBs), enables generation of geometries that are more closely aligned with the articulation solutions proposed by the structure generator, as demonstrated in Table 2. The synergy between conditional graph generation, and constrained 3D generation based on OBBs results in realistic articulated assets with appropriate dimensions, as shown in Fig. 4A.
>
> **Weakness 2.1** Per the reviewer's suggestions, we've added a side-by-side qualitative comparison with NAP (see general author rebuttal). Figure B in attached PDF shows that our model 1) improves the quality of the geometry (see fan), 2) improves the physical plausibility (see laptop motion), and 3) generates more consistent and realistic shapes (see both globes).
>
> **Weakness 2.2** We acknowledge the concern raised regarding the fairness of the comparison, and will clarify this in the revised manuscript. Our aim was to demonstrate the conditioning capabilities of our approach, which are not present in NAP. We do not see a feasible alternative that would ensure a fair comparison and are very open to a fairer comparison that can highlight MIDGArD's capability of conditioning (compared to the lack of any text- or image-based conditioning in NAP) as one of the novelties and contributions of our work.
>
> Regarding the suggestion that NAP should have been given the latent code extracted from the ground truth part geometry: This would create an unfair advantage over our method, which relies solely on single-view images. Additionally, the practical applications where part-geometry is readily available are quite limited, thus the utility of such a capability is uncertain. Leaving this drawback aside, for a fair qualitative comparison one can consider the Part2Motion experiment in NAP (\cite{lei2023nap} Fig 5) as the counterpart to our results for text- and image-based conditional generation provided in the attached PDF Figure A.
>
> **Weakness 2.3** See general response for experiments and results.
>
> **Question 1** By "consistency across the graph," we refer to the semantic coherence of part relationships within the articulated structure. For instance, a "bottle" is unlikely to contain a "drawer". We found that adding categorical information to the graph improves the consistency. In MIDGArD-generated data, we found that our model generates 100% valid examples (see general author rebuttal).
>
> Qualitatively, this advantage of our approach is seen in the failure cases (attached PDF Fig. C). In NAP, the lack of categorical information leads to objects of mixed parts that do not fit together. In our approach, there may be issues with the geometry or kinematics, but the object parts remain consistent.
>
> **Question 2**
> A fair comparison with CAGE is not possible, because CAGE was designed for a very limited setting, i.e., (1) providing shape geometry via \textit{part retrieval} instead of part generation. This method does not generalize and leads to failure cases where parts overlap and don't fit (see \cite{liu2023cage} Fig 10), (2) assuming canonical poses of the objects and axis-aligned bounding boxes. Thus, the experiments in CAGE are restricted to 8 categories from PartnetMobility (3) requiring actionable parts, e.g., doorknobs and handles, which were added by the authors of CAGE and only for 8 categories.
>
> **Question 3** We have corrected and clarified the mentioned line. Indeed, for NAP we refer to the latent encoding the geometry. For CAGE, it is rather the part-retrieval procedure that prevents control on this level.
> Our framework allows users to specify part-level details through images and text, e.g., requiring a lamp to consist of a "lamp stand" and a "lamp shade" and guiding the style of the lamp shade with an image. This control mechanism is not available in prior work.
>
> **Question 4** It generalises. See general response for results.
>
> **Limitation 1** One could leverage existing image segmentation pipelines to automatically extract relevant images from a single picture. Also, one image oftentimes suffices (e.g. sunglasses), 3) we are planning to provide a library of part images to assist guidance.
>
> **Limitation 2** As shown in the attached PDF Figure A, the user can directly input images and text instead of post-editing.

---

> > ### Comment · Reviewer_T3Uj · 2024-08-11
> >
> > I appreciate the effort and detailed response provided in the rebuttal.
> >
> > At a high level, I agree with the authors that modularity is an effective strategy for enabling flexible control over part geometry. Although the current approach—providing images of each part rendered separately—feels somewhat unnatural, this work is trying to address an important problem and has the potential to inspire further exploration in this increasingly significant field. I appreciate the insights and the effort that went into this paper.
> >
> > However, I still have concerns that the experiments in the current state cannot well support the central claims and effectiveness of the proposed components.
> > - **About controllability**: I have concerns about the practicality and effectiveness of the proposed approach based on the limited results presented.
> >     - For image control: It would be more convincing to show qualitative results to demonstrate how different image inputs affect the final output in variations while keeping other attributes fixed. Plus, the requirement of per-part rendering may limit the method's generalizability to real-world scenarios. While the authors suggest that these images could be extracted from a single picture using image segmentation in the rebuttal, there is no supporting evidence for this claim, making the viability of this approach uncertain.
> >     - For text control: The results provided do not sufficiently demonstrate cases where each part's geometry is edited using purely text-based inputs. The supplementary material only includes two examples of combined image and text control, with four additional examples in the rebuttal PDF. It remains unclear how the text input influences the process, and how diverse and compatible outputs can be generated with varying descriptions.
> >     - For graph control: there is no evidence presented on how graph-based control can be implemented or how effective and useful it might be.
> > - **About representation** used in the structure generator: this work introduces several modifications to the representation used in NAP. However, I consider only two of them (using OBB and image latent) to be critical and well-supported contributions. Denoising joint parameters on the Plucker manifold is an interesting design, but the work lacks ablation studies or evidence to demonstrate its specific advantages.
> > - **About the claim of “more natural joint motion” and “less physical constraint violation”**: Only three cases are presented for qualitative comparison with NAP—one in Fig. 3 of the main paper and the laptop and drawer examples in the rebuttal. This limited number of examples is insufficient to demonstrate improvement across a broader distribution, as it is unclear whether these cases were selectively chosen.
> > - **Comparison with CAGE**: I disagree with the authors' assertion that this comparison is impossible. I believe it is a crucial comparison that should be conducted, at least qualitatively.
> >     - I understand that CAGE makes certain simplifications in its assumptions, which may work well for specific categories of objects. However, these categories represent only a subset of the objects considered in MIDGArD. It should be feasible to demonstrate certain aspects of the comparison for these categories.
> >     - On the geometry side, both NAP and CAGE can operate in a retrieval-based setting. My understanding is that it should be an alternative mode for MIDGArD, using an image as a retrieval proxy. In this context, it is important to compare both the retrieval and generation modes of MIDGArD with NAP and CAGE. The difference between generation and retrieval modes does not justify excluding such a comparison.
> >
> > Overall, I believe this is a valuable work in terms of its motivation and potential impact. Once the experiments are fully completed to substantiate all the contributions the authors claim, I would be inclined to support the acceptance of this paper. However, in its current form, I would suggest resubmission in a future round.

---

> > > ### Author Response · Authors · 2024-08-13
> > >
> > > We appreciate your detailed feedback and your recognition of our key contributions, which you have deemed "critical and well-supported." While we regret any oversight in our previous analysis, we would like to clarify that our review does indeed address most concerns you raised. It is possible that these points may not have been fully apparent, and we welcome this opportunity to elaborate further:
> > >
> > > - **Controllability - Image control:** The qualitative impact of varying image inputs while keeping other attributes fixed was precisely illustrated in Fig. 3 of the main paper as well as Fig. 5 of the supplementary material, and Fig. D of the rebuttal. These examples demonstrate the modularity and flexibility of our approach in controlling part geometries.
> > > We respectfully disagree with the impracticality claim of part image condition, considering that (1) images are easier to handle than 3D representations, (2) our experiments show the shape generator works even without input image condition (the results will be integrated into the revised version of the paper) and (3) modern segmentation methods can streamline the process (but falls outside the scope of this paper).
> > >
> > > - **Controllability - Text control:** In addition to the text+image conditioning examples provided in our supplementary material, we have conducted further experiments where the shape generator is conditioned solely on textual descriptions. Our findings indicate that removing image conditioning does not degrade the quality of the generated objects. Variations in text label (e.g., changing ''furniture'' to ''oven'') leads to corresponding changes in geometry.
> > >
> > > - **Controllability - Graph control:** We regret any confusion regarding graph conditioning. The term graph conditioning refers in the original manuscript to the possibility of using any feature of the graph object produced by the structure generator, as an additional condition mechanism to the shape generation process. During our experiments, we only used the node bounding box features, as stated in line 332-333 of the article. Combined with our bounding box prior, this design choice is a core innovation that significantly improves the physical plausibility of generated objects, as evidenced by the results in Table 2.
> > >
> > > - **Representation:** Regarding denoising on the Plücker manifold: we only claim that it "enhances interpretability" (line 201 in the manuscript) and "eliminating the necessity for iterative projections" (line 212). These claims are supported *by definition*, since our method indeed eliminates the need for post-processing and makes the output directly interpretable as Plücker coordinates (see methods section). These benefits are inherent to the design and are thus supported by the methodology itself, requiring no additional empirical validation. As part of our ablation study, we evaluated the ID metrics—MMD, COV, and 1-NNA—comparing unconditional generation scenarios with and without Plücker manifold parameterization. The results are presented in the table below:
> > >
> > > |  Metric \ Method | Ours no-manifold  | Ours + manifold  | NAP |
> > > |-----|-----|-----|-----|
> > > | 1-NNA | 0.6221 | **0.5831** | **0.5831** |
> > > | MMD | 0.0270 | **0.0264** | 0.0282 |
> > > | COV | 0.4779 | **0.4857** | 0.4675 |
> > >
> > > These results suggest improvements in terms of coverage and MMD, supporting our claim that diffusion over Plucker manifold improves asset consistency.
> > >
> > > - **About the claim of ''more natural joint motion'' and ''less physical constraint violation'':** This was covered by experiment 3) and 4) in our general Author Rebuttal above, which provides *quantitative* evidence for these points.
> > >
> > > - **Comparison with CAGE:** We agree that a comparison with CAGE on a limited set of categories and with geometry retrieval could be feasible; however, our modular approach is specifically designed to enable fine-grained control and flexibility in part generation, which contrasts with CAGE’s retrieval-based method. Implementing a retrieval-based mode within our framework would necessitate significant modifications that would undermine one of our method’s key strengths—its ability to generate parts from scratch using multimodal inputs. We believe that such a comparison, would not fairly represent the advantages of our method in terms of generalizability and user control. For the sake of transparency, we will explain in the revision why we did not compare with CAGE directly.
> > >
> > > While we are currently unable to include additional figures due to rebuttal constraints, we will incorporate the results into the revised manuscript. We hope this response addresses your concerns and clarifies the contributions of our work. We are committed to further improving our manuscript based on your valuable feedback and are confident that the additional evidence we plan to include will more clearly demonstrate the effectiveness and potential impact of our approach.
> > >
> > > Thank you again for your constructive feedback.

---

> > > > ### Comment · Reviewer_T3Uj · 2024-08-13
> > > >
> > > > Thank the authors for the further discussion and clarification.
> > > >
> > > > Your response did address some of my concerns, so I can raise my score to weak reject for all the additional effort made during the rebuttal. I believe this paper can be much improved to demonstrate its contribution more clearly and precisely by refining the manuscript and adding more experiment results.
> > > >
> > > > However, I cannot base my final decision on the future revision of the manuscript, so I would still suggest the next round of submission.
> > > >
> > > > Just a clarification of my suggestions:
> > > > - Regarding "qualitative results to demonstrate how different image inputs affect the final output in variations", I was suggesting showing multiple outputs by giving the same set of image inputs (with different input examples of course). From a user perspective of a generative model, it would be interesting to see diverse outputs in terms of bounding box layout and kinematics under a part shape constraint.
> > > > - Regarding the comparison with CAGE, I think both the retrieval and generation mode of MIDGArD in comparison with prior work can help to show the advantage in fine-grained control in several aspects (e.g., semantic consistency given object category, geometry/motion compatibility given different levels of part description, etc.) and also reveal the limitation of prior work. I would expect the generation mode can produce more faithful geometries compared with retrieval-based methods, which demonstrate the necessity of the shape generator module proposed in this work.

---

> > > > > ### Author Response · Authors · 2024-08-14
> > > > >
> > > > > Thank you for acknowledging the additional effort and enhancements made. We would like to assure you that we will incorporate the experiments and improvements into our manuscript, including more qualitative results for image control as suggested.
> > > > >
> > > > > We would appreciate if you could also transfer the updated score to your original response (if possible).

---

### Author Rebuttal · Authors · 2024-08-07

We would like to thank the editors and reviewers for the time they devoted to reviewing our paper and for their valuable feedback and constructive criticism. We have endeavored to address every suggestion and additional comment to the best of our abilities. Below, we provide a summary of our approach to the review:
*  **Additional Experiments and Comparative Analysis:** We provided additional experiment results, such as a (1) qualitative and quantitative side-by-side comparison to NAP [37], (2) additional conditional generation results, and (3) quantitative analysis of the plausibility and consistency of the generated assets (see below).
* **Clarifications and Revisions:** We have improved the clarity of our method and results thanks to thoughtful feedback of the reviewers and thoroughly incorporated all reviewer's feedback. Detailed descriptions of the data processing in the structure and shape generator are provided, along with empirical results highlighting conditional generation capability and qualitative improvements.
* **Related Work:**
    We have revised the related work section to better contextualize our contributions within the existing literature and emphasize the novelty and impact of our work.

We hope these revisions and clarifications address the reviewers' concerns and enhance the understanding of our contributions. We remain committed to further improving our manuscript and welcome any additional feedback.

Best regards,

Authors of the submitted paper

**------------------- Additional experiments------------------**

**1) Conditional generation capability**
We provide additional results demonstrating the conditional generation capability of our approach in a "PartToMotion" setup where the model is provided with part features only (i.e. no joint data) and outputs consistent articulated assets (see attached PDF - Figure "conditional generation"). In contrast to NAP, our model can be guided with image and text input instead of requiring full geometries.

**2) Side-by-side comparison**
Figure B in the attached PDF provides a side-by-side comparison between our approach and NAP. Since the graphs generated by NAP do not include categorical information, we manually go through the generated data of NAP and MIDGArD and selected pairs of assets with the most similar appearance. Figure B shows that our approach 1) improves the quality of the geometry (see the fan and holes in drawer-meshes), 2) improves the physical plausibility (see laptop motion), and 3) generates more consistent and realistic shapes (see both globes).

**3) Physical plausibility**
We provide further results supporting the improvement in physical plausibility achieved with our method. In Figure 4, we have already shown qualitative examples where NAP yields unrealistic joint ranges. Our method alleviates such failure cases by introducing categorical joint labels in the articulation graph. Here, we support this observation with a quantitative analysis of the generated joint types. Specifically, we compare the distribution of joint types in the training data with those in samples generated by NAP and MIDGArD (400 samples each). To minimize the impact of objects having many joints of the same type, we count each joint type only once per object.

|                     | Screw | Revolute | Prismatic |
|---------------------|-------|----------|-----------|
| Training data       | 0.06  | 0.62     | 0.32      |
| NAP-generated       | 0.95  | 0.01     | 0.04      |
| MIDGArD-generated   | 0.02  | 0.62     | 0.36      |

The results indicate that NAP predominantly produces screw joints, despite their low occurrence in the training data. Conversely, the objects generated by MIDGArD exhibit a joint type distribution similar to that of the training data, thereby enhancing the physical plausibility of the generated data. The Chi-Square statistic, which measures how much the observed counts deviate from the expected counts, confirms the large difference with $\chi^2(\text{NAP}) = 5618$ whereas $\chi^2(\text{MIDGArD})=12.7$. NAP's high $\chi^2$ value indicates that there is a large difference between the observed and expected counts, whereas MIDGArD's low $\chi^2$ indicates that the observed data are close to the expected data. More intuitively, the maximal misalignment of our method in the three joint categories is 4% compared to maximal mismatch of 89% for NAP.

**4) Consistency of parts within the object**

By "consistency" we refer to the semantic coherence of part relationships within the articulated structure. For instance, a "bottle" asset is unlikely to contain a "drawer" body. We found that our approach of adding categorical information to the graph improves the consistency. Unfortunately, it is not possible to measure the consistency within NAP-generated objects due to the lack of categorical information. In MIDGArD-generated data, we can compute whether part-types (e.g., "leg") occur in conjunction with a fitting asset type (e.g., "table"), or whether they are mixed (e.g., a "leg" being part of a "laptop"). Assume that every part-asset-type combination occurring in the training dataset is valid, our model actually generates 100% valid examples. In other words, none of the generated graphs is a mixture of parts that typically belong to different asset types.

Qualitatively, this advantage of our approach is visible in the failure cases (see attached PDF - Figure C). In NAP, the lack of categorical information leads to objects of mixed parts that do not fit together. In our approach, there may be issues with the geometry or kinematics, but the object parts remain consistent.

**5) Generalisation to images from various perspectives**: We show in the attached PDF - Figure D that our conditional part-generation generalizes to images from different viewpoints.

These results will be included in the revised manuscript.

---

### Decision · Program_Chairs · 2024-09-25

**Decision:**

Accept (poster)

**Comment:**

This paper received divergent initial opinions with three reviewers in support of acceptance and one negative reviewer.  The main concerns expressed by the reviewers were: unclear comparisons or missing comparisons against the most relevant prior work (NAP and CAGE), questions about the fairness of the evaluation methodology, a somewhat contrived "single part in an image" conditioning setup, seemingly marginal quantitative improvements over the prior work baselines, and a lack of discussion about how the contributions of the paper relate to those of the most relevant prior work (NAP and CAGE).

The rebuttal responded to reviewer concerns with detailed clarifications and additional experiments.  The positive reviewers remained positive. The one initially negative reviewer raised their strong reject to a weak reject but remained unconvinced with the rebuttal clarifications on whether the comparisons and evaluation against prior work are complete and fair enough to substantiate the claimed contributions.  Due to the severity of these concerns, there was extensive discussion in the rebuttal and post-rebuttal period.  Taking into account the willingness of the authors to clarify and improve the paper, the AC and reviewers support acceptance, provided the following revisions are incorporated into the manuscript:
1. More precisely articulate the contributions (e.g., how image/text/graph control is specified by the users, and how the Plücker manifold enhances the representation).
1. Acknowledge aspects of the approach inspired by prior work and how their benefits align with those demonstrated in the prior work (e.g., conditioning on object categories to achieve semantically consistent parts and adding joint type as a categorical label to improve kinematic plausibility, as shown in CAGE).
1. Qualitatively demonstrate the diversity of outputs when using the same image/text input.
1. Be transparent about how the comparison with NAP is conducted and clarify why CAGE was not included as a comparison point.

Thus, the AC recommends acceptance and strongly encourages the authors to incorporate clarifications and revisions from the review cycle and especially targeting the above points, into the final manuscript.